neuroscience, computational biology, theoretical biology

neural computation, neural networks, state estimation, granule cell layer, sensorimotor

**Author for correspondence:**
Ensor Rafael Palacios
e-mail: ensorrafael.palacios@bristol.ac.uk

†These authors contributed equally to this study.

# Accounting for uncertainty: inhibition for neural inference in the cerebellum

Ensor Rafael Palacios[1], Conor Houghton[2,†] and Paul Chadderton[1,†]

[1]School of Physiology Pharmacology and Neuroscience, University of Bristol, Bristol BS8 1TH, UK
[2]School of Computer Science, University of Bristol, Bristol BS8 1UB, UK

ERP, 0000-0001-8170-3397; CH, 0000-0001-5017-9473; PC, 0000-0002-6276-1011

Sensorimotor coordination is thought to rely on cerebellar-based internal models for state estimation, but the underlying neural mechanisms and specific contribution of the cerebellar components is unknown. A central aspect of any inferential process is the representation of uncertainty or conversely precision characterizing the ensuing estimates. Here, we discuss the possible contribution of inhibition to the encoding of precision of neural representations in the granular layer of the cerebellar cortex. Within this layer, Golgi cells influence excitatory granule cells, and their action is critical in shaping information transmission downstream to Purkinje cells. In this review, we equate the ensuing excitation–inhibition balance in the granular layer with the outcome of a precision-weighted inferential process, and highlight the physiological characteristics of Golgi cell inhibition that are consistent with such computations.

## 1. Introduction

Sensorimotor coordination or control can be regarded as the realization of expected sensation via movement. It involves interactions between an agent and its environment; like when a mouse is actively gathering information with its whiskers. In order to control these interactions, the brain must be able to approximate or predict the consequences of forthcoming action. This relies on accurate estimates of behaviourally relevant states (such as whisker position) generated by an underlying model of how states relate to one another. Estimates are intrinsically uncertain, reflecting stochasticity in sensory channels and dynamics of states. Hence, when considering the neural implementation of an estimation process, it is desirable that neural circuits are capable of representing estimates conditioned on their associated uncertainty; in other words, the underlying models ought to be probabilistic.

The cerebellum has long been posited to instantiate probabilistic internal models for estimation of rapidly varying external states [1], whether somatic, such as limb kinematics [2], or environmental, for example, moving targets [3]. In the cerebellum, these models are deemed to support sensorimotor control [4–6], as well as more abstract mental representations [7], by complementing ongoing neural computations in other brain regions with internally generated, delay-free probabilistic estimates of stochastic external dynamics, built upon past experience and integrating multiple sources of noisy information.

These ideas are long-standing, but it remains unresolved how various components of the cerebellum could specifically contribute to state estimation. In general, only activity and plasticity of Purkinje cells, the output of the cerebellar cortex, have been associated with this computation; however, inferential processes occur all the way through the hierarchy of an internal probabilistic model. Here we consider the first step in cerebellar cortical state estimation, by proposing a role for inhibition in the granular layer. This network, comprising about half of the neurons in the mammalian nervous system, relays all extracerebellar input that is directed via mossy fibres (MFs) to Purkinje cells [8–10] (figure 1). The granular layer is made up of excitatory granule cells and inhibitory Golgi cells, and interactions between these two neuronal

**Figure 1.** Information flow through the cerebellum. Information from extracerebellar structures enters the cerebellum via MFs (violet arrows) and climbing fibres (not shown). MFs contact both the cerebellar cortex and nuclei; in the former, they synapse in the granular layer, the first stage of information processing along this pathway. Here reside densely packed excitatory granule cells (in red), and sparsely distributed inhibitory Golgi cells (in blue), which inhibit vast and overlapping groups of granule cells. Interaction between these two neuronal populations determine how information is transmitted downstream to Purkinje cells (in orange) through granule cell ascending axons and parallel fibres (red fibres). Purkinje cells generate the sole output of the cerebellar cortex, influencing neural activity in the cerebellar nuclei (brown box). Nuclear neurons depart axons back to extracerebellar structures (brown arrows), also making collaterals that terminate as MFs in the granular layer. The dotted box inscribes all MF terminals in the cerebellum.

populations determine network responses to external (MF) perturbations. Two aspects are key to understanding neural dynamics within this network: firstly, granule cells are numerous but individually receive only a small number of inputs (four excitatory and four inhibitory connections each on average [11,12]); secondly, Golgi cells are sparse relative to granule cells, but each neuron contacts hundreds to thousands of granule cells through an extended axonal arborization. Thus, Golgi cell inhibition is likely to have a big impact on individual granule cell activity and putative inferential processes in the cerebellum.

We explore this possibility by discussing the role of Golgi cell inhibition in the context of state estimation in the cerebellar cortex, and set out a link between high-level theoretical descriptions of cerebellar computations and their neural substrates. We start from the assumption that neuronal activity encodes estimates or predictions of somatic and environmental states that enable guidance, coordination and refinement of action; then, we argue that Golgi cell inhibition promotes accurate state estimation, by adaptively tuning excitatory responses encoding those estimates. Crucially, such tuning rests upon (neural mechanisms signalling) the *precision* of the information driving the inferential process, so that the excitation–inhibition balance in granule cell populations becomes the result of a precision-weighting process. This view ultimately affords a new interpretation of observed inhibitory mechanisms in the granular layer.

## 2. Precision in state estimation

Many aspects of brain functioning can be phrased in terms of probabilistic inference and learning processes [13–15]. In this framework, inference and learning are based on probabilistic models entertained by the brain, representing somatic and environmental variables or states, their dynamical interactions and link to sensory input [16]. Central to this argument is the notion of uncertainty, describing the spread or variance of belief distributions assumed to be implicitly encoded by neural activity. Whatever the exact form of this encoding, one can argue that the variance of the implicit distributions depends in the first place on the quality of data available to the network. In other words, uncertainty represented in neural activity should be a function of input precision, a measure of the reliability of input that determines how much this drives belief updating (box 1).

In biological neural networks, precision naturally translates into population gain [26], which scales or weights presynaptic input and adjusts its capacity to elicit voltage changes in the target population. The underlying idea is that a neural circuit is a system with endogenous or autonomous dynamics, whose activity is not entirely determined by external stimuli; its response to events can contextually vary, conditioned on their precision. Here we assume that inputs reporting more precise representations are associated with higher population gain, that is, a stronger impact on downstream network dynamics—whose output in turn is implicitly linked to more precise distributions.

This brings us to two key points: first, the quality or precision of information is not reducible to its content, meaning that neural mechanisms signalling *what* is represented can be different from those signalling *how* it should be represented. For instance, the identity and activity pattern of upstream neurons can be related to the nature of a stimulus encoded, whereas the postsynaptic gain to the amount of information

**Box 1.** Input precision changes neural response.

When investigating the functions of a neural network, we usually try to identify which features of the body or world are encoded in the activity of its constituent neurons. Underlying this approach is the assumption that there is a mapping between the activity of the network and the outer states. Because this mapping is indirect—mediated by vicarious input about the system—it licenses an interpretation of neural circuits as internal models inferring causes of their input, like a patch of V1 reflecting the possible presence of a luminous bar projected onto the visual field. Importantly, this mapping is necessarily probabilistic, because the dynamics and interactions between states and sensory signals are noisy. By accounting for this stochasticity, neuronal activity comes to reflect probability distributions over states.

The cerebellum is thought to instantiate internal models for motor and cognitive calibration and adaptation. Neural activity in this region has indeed been observed to accurately encode dynamics of somatic or environmental states, such as whisker position in the mouse cerebellum [17]. These representations in turn contribute to sensorimotor coordination by refining motion [18] and sustaining or altering neural activity in other brain regions, such as the neocortex [19–22] (panel *a*).

In order for network dynamics in the cerebellum to reflect inferential processes, it is necessary that uncertainty in state estimation influences neural activity. There exist different models of how probability distributions can be encoded by neural populations (see [23] for an example in the cerebellum), some of which highlight the possibility that neural activity scales with the precision of the encoded distribution [24], while becoming sparser as an effect of divisive normalization [25] (panel *b*). Notably, activity levels in recipient populations result from a combination of input and population gain/ responsiveness, which here we associate with input precision. In other words, we argue that (un)certainty in neural representations is determined by the quality of information in the input driving those representations (panel *c*). Notice that input precision-weighting relies on the capacity of the network to assess this precision. Here we address this possibility and propose that Golgi cells in the cerebellar granular layer mediate the link between neural dynamics and state estimation, by making network excitability sensitive to and reflective of uncertainty in inferential processes.

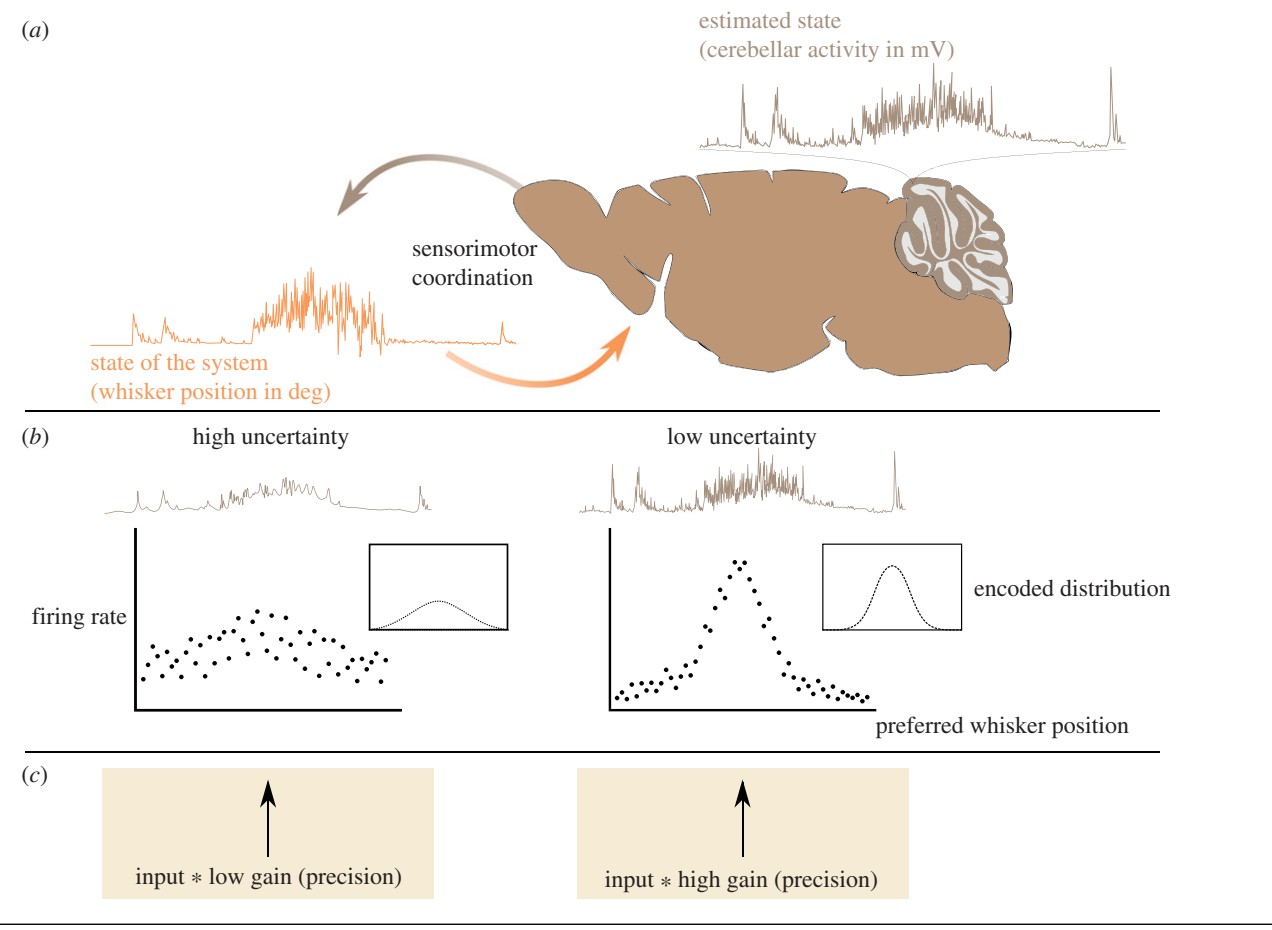

transmitted. Second, the precision of an input, realized as population gain and ultimately translated into patterns of excitation–inhibition balance, is related to the concurrent behavioural context. This can be exemplified through attentional gain modulation in visual and auditory cortex [26–28], where contextualization (weighting) of sensory stimuli by their precision can be accomplished via temporal

coincidence of pre and postsynaptic activity, increasing probability of conversion of pre to postsynaptic spikes. In this case, gain-by-synchrony depends on both bottom-up (e.g. intrinsic saliency of the stimulus) and top-down (attentional) effects [29–31], which are a function of behaviour. It follows that the extent to which a stimulus can be relevant for ongoing inference—under a certain behaviour—translates

into precision-weighting of that stimulus via gain modulation.

The cerebellar cortex receives input via MFs from virtually every part of the brain. This input is rich, encompassing multiple sensory and motor modalities [32–36] as well as cognitive domains [37]. Moreover, its nature can be both predictive (e.g. anticipatory reward-related signals) and postdictive (e.g. sensory feedback) [38–40], encompassing the entire period of movement execution (e.g. [39]). Consequently, at any given time a huge amount of information can potentially be transmitted to the granular layer via MFs. However, only a fraction of this information is likely to be relevant at any given moment in time; this fact intimates the necessity for the cerebellar cortex to select or prioritize some and not other sources of input, so that only information that is relevant in a particular behavioural context can affect state estimation. For example, while engaged in a visuomotor task, postsynaptic responses to MFs conveying confounding auditory signals might be dampened. With respect to the cerebellar cortex, precision encoded in various extracerebellar regions must be translated and implemented in a common way within the granule cell population, in a manner which is instrumental for state estimation, that is, causing downstream layers to appropriately respond to encoded precision. Accordingly, inhibition in the input layer of the cerebellum appears capable of operating these fundamental operations.

## 3. Golgi cells underlie precise granular layer computations

The inhibitory network in the cerebellar granular layer is simpler than in cerebral cortical regions, lacking cellular diversity and extremely complicated intra- and inter-areal top-down feedback modulation (cf. [41,42]); nonetheless, it is suited to effectively balance excitation in granule cells. Golgi cells act through both a hyperpolarizing current that lowers granule cell resting potential—efficiently thresholding or gating MF input—and through an increase in membrane conductance or shunting inhibition—associated with faster membrane dynamics and ultimately a biased sensitivity towards synchronous presynaptic activity [43,44]. As a result, Golgi cells can set the excitability or responsiveness of granule cells, approximated by the operative point (position and slope) of their F-I (frequency-current) curve, controlling propagation of MF activity within the cerebellar circuit. From a neural inference perspective, this propagation should be conditional upon the precision of information transmitted, implying that Golgi cell inhibition is sensitive to signals that are most relevant for present belief updating. It is therefore necessary to identify which mechanisms may inform granule cell excitability via inhibition in this context-dependent manner. One distinction mentioned above is between bottom-up and top-down sources of conditioning; beyond this, various biophysical mechanisms might determine how Golgi cells operate. In this section, we highlight those mechanisms that may allow Golgi cell inhibition to perform precision-weighting of the input. First, we see how inhibition sets neural gain to match average levels of activity. Then, we consider time-varying inhibition and its modulation by temporal and spatial properties of the input; in doing so, we characterize the temporal unfolding of MF activity and its spatial organization as a proxy for its intrinsic (bottom-up) precision. Finally, we address

mechanisms, such as modulation of Golgi cells by neuromodulators or nucleocortical projections, that do not directly depend or arise from current MF input, yet control how these are transmitted by changing the endogenous state of the granular layer; we refer to these as top-down mechanisms signalling expected precision of the input.

A substantial component of inhibition is tonic, hinging on constantly activated extrasynaptic receptors that are responsive to ambient levels of neurotransmitter concentration [45]. This persistent form of inhibition, arising in part from non-vesicular sources of γ-aminobutyric acid (GABA) [46,47], is favoured by the synaptic organization of the granular layer. Most if not all synaptic connections to granule cells are indeed located in special structures called glomeruli, which form isolated microenvironments where neurotransmitter (both GABA and glutamate) can accumulate and easily diffuse [12,48–50]. In these compartments, ambient concentrations of GABA are sufficient to persistently activate high-affinity $\alpha_6\delta$-subunit containing GABAA receptors [51]. *In vivo*, tonic inhibition minimizes granule cell responsiveness to uncorrelated, temporally scattered inputs [52], while maintaining an exquisite sensitivity to salient (e.g. sensory-evoked) stimuli [53]. Therefore, tonic inhibition appropriately fixes granule cell excitability to match the average levels of MF activity, establishing a slowly changing threshold on neural gain discriminating noise from signals. In mathematical terms, this may be equivalent to a prior over expected precision of the input required for its propagation. At a behavioural level, loss of motor coordination resulting from the disruption of tonic inhibition, for example, owing to alcohol consumption [54], might then reflect global alterations in representational uncertainty.

On top of a persistent inhibitory conductance, feedforward and feedback synaptic loops enable Golgi cells to dynamically modulate granule cells by following rapid variations in network activity [55]—although the exact contribution of these loops is still unknown. Phasic inhibition underlies balanced dynamics of excitation and inhibition in granule cells. Notably, phasic inhibition from Golgi cells can promptly track changes in MF spiking behaviour while, at the same time, accumulate over Golgi cell spike trains to match input firing rates [50,56–58]. The ensuing coordination of excitation and inhibition, on a timescale ranging from few to hundreds of milliseconds, can determine which input patterns elicit responses based on the evoked instantaneous balance. Accordingly, when inhibition is temporally matched to excitation, granule cell firing is reduced but becomes more similar across cells [59]: *in vivo*, this could favour, for example, selective transmission of the synchronous and invariant component of MF stimuli to Purkinje cells, by virtue of its stronger impact on postsynaptic neurons. Moreover, inhibition can preserve temporal information in granule cell output by rapidly trailing excitation and forcing a sharp integration window of couple of milliseconds for excitatory post-synaptic currents [57]. Overall, balanced dynamics could increase the capacity of granule cells to reliably transmit temporally structured information—here associated with high precision representations. This is in agreement with the general observation that the granular layer faithfully encodes extracerebellar activity [60–64]; and resonates with the idea of a precision-weighting mechanism relying on inhibition and sensitive to bottom-up dynamics, such as synchrony in MF input enhancing temporal coordination across subsets of Golgi cells [65].

Along with temporal features of Golgi cell inhibition, the spatial arrangement of Golgi cell processes may also play a role in the contextualization of incoming information [58]. Notably, there is a mismatch between the narrow granular layer region from which Golgi cells receive excitatory inputs (determined by the dendritic tree), and the region extending hundreds of micrometers over which they exert inhibitory influence (determined by the axonal plexus). In the present discussion, lateral inhibition could be linked to representational precision via its effects over correlations among different streams of MF input. Excitation–inhibition balance at any location in the granular layer could then reflect—via horizontal mixing of Golgi cell signals—the precision of the local information, relative to its surround. In practice, this could lead to an increase of fast correlations among clusters of granule cells that are excited by common MFs, and a simultaneous decrease of slower correlations across competing patches of granular layer—replicating observations in structures that share a similar geometry, like the olfactory bulb [66].

This contextual modulation of granule cell excitability relies on spatial constraints of information driving Golgi and granule cell populations, which in turn depend on different anatomical properties of the network. MFs show substantial anisotropic divergence in the granular layer [34,67], which enables integration of various sources of information at the level of single granule cells, but prevents the emergence of ordered, neocortical-like receptive fields. As a consequence, fast correlations among Golgi cells (and inhibited clusters of granule cells) sharing MF input might be more evident within distributed, scattered groups of cells [68].

Another important anatomical property is the presence of millimetre-long granule-Golgi cell connections mediated by parallel fibres [69]. Parallel fibres have been linked to extended oscillations in the granular layer during rest [70], possibly setting a global pace for network computations and dynamics. Notably, these connections appear to be qualitatively different from local contacts made by ascending granule cell axons onto Golgi cells, which resemble more the faster and stronger MF-Golgi cell synapses [55,71]. It follows that upon localized activation of MF terminals, parallel fibres might preferentially contribute to slow correlation of granule cells across the transverse axis [72], while ascending axons precisely entrain spiking of surrounding Golgi cells. Furthermore, the existence of electrical connections among Golgi cells further increases their sensitivity to temporal coincidence of local excitation, enhancing synchrony or alternatively asynchrony in and between granule cell clusters [65,73,74]. Therefore, different degrees of correlations might coexist in the granular layer, following properties of MF input and connectivity structure within the network, which might result in balanced dynamics of excitation and inhibition reflecting the statistics (precision) of information encoded.

Finally, precision-weighting for state estimation does not depend solely on properties intrinsic to the inputs, but also on selective mechanisms modulating states of the network. Analogously, Golgi cells are both driven by the same MF inputs that elicit activity in granule cells, and are influenced by neural components located within or external to the cerebellum. *In vivo*, the granular layer is characterized by endogenous activity owing to spontaneous firing of MFs and Golgi cells [62,70,75]; this autonomous state affects the evoked response elicited by a stimulus, and is itself under the control of various mechanisms. In particular, within the cerebellar cortex, climbing fibres, Lugaro cells and Purkinje cells all directly or indirectly modulate Golgi cell activity. [76–78]. From cerebellar nuclei instead, excitatory neurons give rise to MF collaterals innervating glomeruli [79], and inhibitory neurons selectively contact Golgi cells through long-range axons [80]. Moreover, Golgi cells are also sensitive to a variety of neuromodulators including serotonin [81] and noradrenaline [82], which exert opposing actions upon granular layer excitability. Clearly, these sources of input exert very different effects on information processing, and their specific role is still unresolved; nevertheless, this intricate circuit highlights the importance of properly tuning inhibition in the granular layer in order to contextualize incoming information. This is central for putative state estimation in the cerebellar cortex, as it depends not only on current local observations, but also on past inference, systemwise states and coordination with other brain structures.

In conclusion, there appear to be a variety of mechanisms that could inform the granular layer about precision of MF input, irrespective of the extremely diversified nature of those inputs. These mechanisms condition granule cell excitation through Golgi cell inhibition, which constitutes the unique local feedback of the network. In this sense, Golgi cells emerge as a crucial hub for precise state estimation in the cerebellar cortex (figure 2).

# 4. Discussion

We have considered how putative state estimation in the cerebellar cortex relies on appropriately tuning of neural dynamics within the granular layer. In this picture, Golgi cell inhibition underlies selective responsiveness or gain of subsets of granule cell population to extracerebellar activity, controlling to what extent the latter drives state estimation. This is a form of input precision-weighting, and depends on information about precision being accessible to the network. Accordingly, several mechanisms related to Golgi cell functioning could provide this information, and therefore adapt granule cell population gain to input precision. In particular, we interpret the optimal signal-to-noise ratio set by tonic inhibition as a prior over expected signal precision, possibly defined by phylogenetic and ontogenetic processes. On the other hand, rapid changes in the excitation–inhibition balance could dynamically accommodate the relative precision of the excitatory drive as a function of its intrinsic properties and top-down modulations, hence controlling precision encoded in the granular layer output (figure 3). The importance of this balance for state estimation is intimated by the remarkable number of mechanisms capable of tuning it. Together, these mechanisms could underpin rapid but accurate changes in neural representations, which would afford to the cerebellar cortex sufficient temporal resolution to encode body and environmental state dynamics [84], in service of well-timed, predictive computations throughout the brain [85–87].

Arguably, neural dynamics underlying state estimation in the cerebellar cortex should necessarily be sensitive to uncertainty associated with inference, as discussed more generally in the context of cortical functioning [26,27,88]. Cortical control of excitation is much more complex than in the granular layer, hinting at a more sophisticated neural inference.

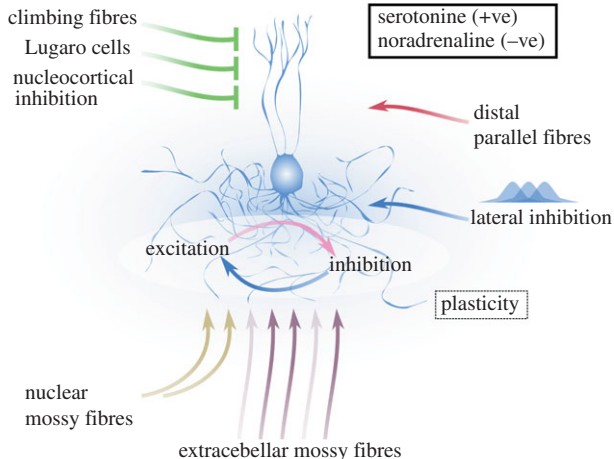

**Figure 2.** Putative mechanisms for precision-weighting control excitation–inhibition balance in the granular layer. Extracerebellar MFs entering the granular layer are presumed to drive state estimation or belief updating in the cerebellar cortex, whose evoked activity encodes precision-weighted estimates of (somatic and environmental) states of the system. This weighting controls the excitation–inhibition balance in granule cells and is induced by activity of Golgi cells. A variety of factors, which we link to input precision, can condition Golgi cells and therefore granular layer dynamics. These include temporal properties (e.g. synchrony) of MF activity as well as its spatial organization, shaping interactions between neighbouring patches of granular layer via lateral inhibition and distal parallel fibres. At the same time, activity from downstream layers and circuits can dynamically adapt the state of the network by changing Golgi cell activity through climbing fibres, Lugaro cells and nucleocortical inhibition. In addition, the serotoninergic and noradrenergic system too can boost and hinder Golgi cell inhibition respectively, possibly promoting systemwise coordination, for example, between neo- and cerebellar cortex. Finally, present states are also influenced by past inference and experience, through nuclear MF feedbacks and plastic changes in granular layer connectivity [83], determining the impact and spatial organization of extracerebellar MFs input. In this picture, Golgi cells are a crucial hub for state estimation in the cerebellar cortex, contextualizing new MF information and conditioning granule cell response accordingly.

Nonetheless, fine-tuning of granule cell activity via Golgi cells also seems to be calibrated by a variety of mechanisms, including top-down signals from cerebellar nuclei, as well as from downstream layers in the cerebellar cortex itself. This in turn should bear on any theory aiming to explain cerebellar computations. In particular, recurrent connectivity within it has usually been neglected or oversimplified; by contrast, a probabilistic inference framework may provide a starting point for explaining this anatomical detail, as shown in this review.

Testing these ideas requires both theoretical and experimental efforts. Here we have assumed a general principle, namely, that precision in neural representations should affect their propagation across the different stages of inference—by tuning population gain. However, future work should aim at investigating the exact nature of this probabilistic encoding throughout the cerebellar circuit. From the experimental side, testing these ideas requires tracking and manipulation of spatiotemporal properties of excitation–inhibition balance in the granular layer, as MF input is transformed into parallel fibre output. The exact shape of ensuing activity patterns depends on many factors, including kinetics variability at the Golgi-granule cell synapses [56], which might result from plastic mechanisms in the granular layer [83]. Previous works have examined

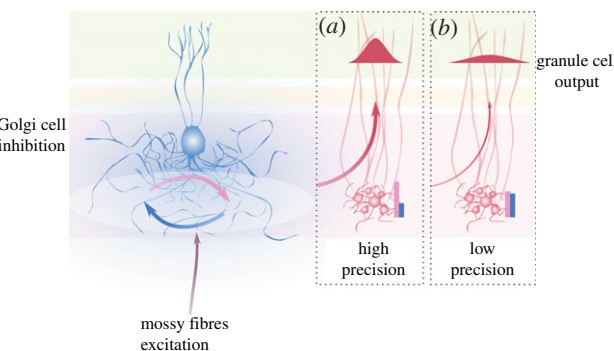

**Figure 3.** Tuning of the system's excitability controls precision of representations. Left: the excitation–inhibition balance in the granular layer (circular blue and pink arrows) is a function of both MF input and neural mechanisms signalling its precision by tuning Golgi cell inhibition. Right: within a population, specific neurons might exhibit higher or lower synaptic gain, depending for instance on stimulus overlap with their receptive field, while at a network level, population gain associated with precision of upstream representations dictates the responsiveness of neural ensembles. Golgi cell inhibition sets population gain, such that the balance of excitation and inhibition in granule cells reflects precision-weighted input; and encodes neural representations whose precision determines their transmission and influence on downstream integrative layers via parallel fibres. In (a), MF input is coupled with high population gain and strongly drives granule cells, pushing excitation (pink bar) to overcome inhibition (blue bar). The ensuing population activity then represents states with high precision, exemplified by the red distribution. In (b), MF input convey less reliable information, and the low gain brings inhibition to balance excitation, making the network almost unresponsive. The small network output, in turn, encodes state estimates with low precision, which will not be effective in driving neural inference downstream.

the consequences of altering excitation levels in this network, showing a direct link to motor impairment, including tremors, ataxia and reduced reflex adaptation [54,89,90]. Interestingly, Golgi cell ablation alters the spatio-temporal patterns of activity in the granule cell population without necessarily producing overexcitation, as the result of compensatory mechanisms such as reduced N-methyl-D-aspartate activity [91]. This highlights the importance of fine-grained granule cell activity patterns for downstream computations [90]—here argued to be the result of Golgi cell-mediated precision-weighting at the first stage of state estimation in the cerebellar cortex. Ultimately, technical advances in the field [92,93] will make it possible to verify or not verify these ideas.

## 5. Conclusion

The cerebellum is deemed to support behaviour through predictive processes, adjusting and refining interactions of the organism with its environment. These processes rely on internal models that capture consistent relationships between states of the system and are probabilistic in nature, taking uncertainty into account. Many data confirm this high-level description of the cerebellum, but it remains to be understood how this can emerge from activity of neural networks. In the present work, we link Golgi cell inhibition in the granular layer with mechanisms sensitive to the relative precision of MF input, and the ensuing excitation–inhibition balance in granule cells with a precision-weighted response.

**Data accessibility.** This article has no additional data.

**Authors' contributions.** All authors contributed to the conception of the study. E.R.P. wrote the manuscript and made the figures, all authors contributed to drafting and revision.

**Competing interests.** We declare we have no competing interests.

**Funding.** This work was supported from the following sources: Wellcome Trust Neural Dynamics PhD studentship to E.R.P.; Wellcome Trust Investigator Award to P.C.

**Acknowledgements.** We thank Laurence Aitchison for feedback on the manuscript.

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
