## [Peer Review File · Proceedings of the Royal Society B: Biological Sciences]

Review History

RSPB-2020-2290.R0 (Original submission)

Review form: Reviewer 1 (Mark Wagner)

Recommendation

Accept with minor revision (please list in comments)

Scientific importance: Is the manuscript an original and important contribution to its field?

Good

General interest: Is the paper of sufficient general interest?

Good

Quality of the paper: Is the overall quality of the paper suitable?

Good

Is the length of the paper justified?

Yes

Should the paper be seen by a specialist statistical reviewer?

No

Do you have any concerns about statistical analyses in this paper? If so, please specify them explicitly in your report.

No

It is a condition of publication that authors make their supporting data, code and materials available - either as supplementary material or hosted in an external repository. Please rate, if applicable, the supporting data on the following criteria.

Is it accessible?

N/A

Is it clear?

N/A

Is it adequate?

N/A

Do you have any ethical concerns with this paper?

No

Comments to the Author

This manuscript reviews helpful material related to the cerebellar Golgi cell - granule cell network and places that material in a probabilistic inference framework. As the authors note, in prevailing theory, the cerebellum is thought to execute forward model prediction. Although prediction requires incorporating uncertainty, the most common formulations of these ideas in cerebellar neuroscience often do not afford much role for uncertainty. Among many other places where uncertainty should propagate through the cerebellar circuitry, the authors here consider its role at the input layer in "weighting" granule cell representations of different information sources via activation patterns of the network of inhibitory Golgi cells.

The topics reviewed are likely to be of use to cerebellar and non-cerebellar researchers, and the probabilistic framework is helpful in a field whose most dominant theories are largely non-probabilistic in nature. The authors also helpfully describe unanswered questions and predictions that may be tested in future research.

I have two general suggestions for the authors' revision.

First, the review's model/hypothesis and language is sometimes too speculative or nonspecific, veering a bit far from "review" territory into theoretical hypothesis territory. To take an example, P10 "tonic inhibition sets an optimal signal-to-noise ratio, acting as a prior over average or expected signal intensity" (as far as I know this is hypothetical, and even as a theory, it's not exactly clear that this solves a necessary uncertainty problem. Aren't there specific uncertainties associated with different modalities?). More generally, the organization of the review sometimes takes the form of a list of known features of Golgi anatomy followed by a guess as to how those features might contribute to probabilistic inference. It would be helpful for the authors to lay out, somewhere near the outset, the set of uncertainties they think are relevant to inference. They can refer to this framework as they make their way through the review of Golgi functional anatomy, rather than springing on readers wholly new probabilistic concepts throughout the paper. Second, in most people's minds, the state prediction relevant to cerebellar processing is done after the input layer, e.g. in the Purkinje cells. This review is framed as addressing the issue of confidence, but it's focused on confidence in the input variables the cerebellum uses for state estimation. I agree that input variance is important to determining confidence in the resulting state estimation. Still it should be more explicitly described to naive readers that this is the first step in the process of inference, and is distinct from computing predictions and the confidence level of those predictions (input confidence must be propagated through to the final distribution of state estimates etc). The rest of these computations are beyond the scope here, but it will help to give readers a concise sense of the overall framework.

Other specific comments:

P3 par2 “In this framework, inference and learning are based on probabilistic models entertained by the brain, representing somatic and environmental variables or states, their dynamical interactions and causal link to sensory input.”

“Causal link” is not a great descriptor. If the task is probabilistic inference, it is typically not relevant whether the connection is causal or correlative.

p.5 par1- “The inhibitory network in the granular layer is simpler than in cerebral cortical regions...”

The authors’ ground their perspective in neocortical top-down inferential inhibitory processing, but to do so fairly they should consider addressing a few things more completely.

First, at the anatomical level there is not really a compelling comparison to be made between the extremely complicated intra- and inter-areal top-down feedback modulation of input to the neocortex, and the very limited recurrent circuitry of the cerebellum. This should give more pause than implied in the present text, and suggests the comparison should be at most a conceptual analog rather than a mechanistic one.

On the other hand, classical cerebellar theories also don’t do justice to what “top-down” processing does exist in the cerebellum (by treating the cerebellum as a feedforward structure without recurrent excitation, in some cases almost as precondition of the computational theory). As the authors note, DCN provides recurrent mossy fiber inputs, and this may indeed be a route for a more sophisticated form of feedback modulation. It might be helpful to stress that this potential mechanism for modulating inputs is a substantial divergence from classical feedforward adaptive filtering theory, and a probabilistic inference framework may provide a starting point for explaining this anatomical detail that is ‘inconvenient’ in other theories.

P6 par1 - In the discussion of Golgi cells’ wide axonal arborization vs narrow dendritic column, I’m not sure I view this as evidence of a lateral inhibition mechanism, at least as presented, nor necessarily as a discrepancy? Granule cell axons are ~1 mm long--perhaps narrow Golgi dendritic columns are sufficient to sample from a granule cell population as widely distributed as that contacted by the diffuse Golgi axonal branches?

P9 Par 3 in the proof opens with a strange semi-repeat of par2

The organization of the last two sections “Adaptive filter model” and “Kalman filter model” is a bit unclear/jarring. Is the purpose of these sections to contrast them with an alternative model in which Golgi cells contribute to Bayesian inference? Is it to explain their shortcomings? Is it to explain that if Golgi cells convey uncertainty, that can be incorporated into these models in order to solve their shortcomings? In addition, it is unfortunate that this section leads directly into the discussion, without any concluding and transitional statements, as it will doubtless confuse readers less familiar with this literature by distracting from the central narrative of the review.

P3 par2 “In neural networks, precision weights presynaptic input...” is asserted as a known truth rather than as a hypothesis (assuming that is the intention? Otherwise needs citations).

A number of assorted typos/missing words/grammatical issues/partial sentences/run-on sentences, please take a look throughout.

Review form: Reviewer 2

Recommendation

Major revision is needed (please make suggestions in comments)

Scientific importance: Is the manuscript an original and important contribution to its field?
Acceptable

General interest: Is the paper of sufficient general interest?
Acceptable

Quality of the paper: Is the overall quality of the paper suitable?
Marginal

Is the length of the paper justified?
Yes

Should the paper be seen by a specialist statistical reviewer?
No

Do you have any concerns about statistical analyses in this paper? If so, please specify them explicitly in your report.
No

It is a condition of publication that authors make their supporting data, code and materials available - either as supplementary material or hosted in an external repository. Please rate, if applicable, the supporting data on the following criteria.

Is it accessible?
N/A

Is it clear?
N/A

Is it adequate?
N/A

Do you have any ethical concerns with this paper?
No

Comments to the Author

This perspective/review advances the hypothesis that the inhibitory mechanisms present in the granule cell layer of the cerebellum could act as a sensorimotor state estimator that encodes uncertainty. The article begins with a rather brief introduction to motor control, probabilistic internal models and state estimation. None of these are sufficiently 'unpacked' for readers unfamiliar with these concepts to understand them. The central question of how the cerebellar cortex might compute state estimation is clearly stated and the proposal to examine the underlying mechanisms is set out. Moreover, the central idea of the cerebellar cortex encoding probability distributions that are used for Bayesian state estimation is interesting. But the authors seem to be unaware that it has been proposed before by Paulin (Paulin et al. 2005. J. Neural Eng. 2 (2005) S219-S234 doi:10.1088/1741-2560/2/3/S06), who has provided compelling arguments for state estimation since the early 1990s (Paulin Human Movement Science 12 (1993) 5-16; Paulin, Brain Behav Evol 1993;41:39-50). Overall, this manuscript provides a good review of the cellular and systems level studies of the cerebellar input layer (excepting those detailed below), but in my view it does not provide a convincing link between the underlying mechanisms and behaviours to the probabilistic state estimation hypothesis. The review thus provides a rather undeveloped description of encoding uncertainty and state estimation and a more complete account of the anatomy and physiology of inhibition the input layer and some descriptions of other theories of cerebellar function. The figures have some artistic merit but convey little scientific content and miss the opportunity to explain the complex concepts discussed. Lastly the manuscript PDF has several major formatting errors. The main text was duplicated twice, with the references and

conclusion missing from the initial copy of the text. Moreover, in the complete second copy there was a duplication of a large block of text in the section on the Kalman filter. In my view the manuscript requires a very major rewrite that includes an accessible explanation of exactly what probabilistic state estimation is, how uncertainty can be used in motor control and incorporates and acknowledges Paulin's and Latham's (see below) work on this. Moreover, the text needs to be much more focussed on the evidence and mechanisms that could support probabilistic state estimation as well as highlighting results that do not fit.

Specific points for consideration

- 1) Expand on explaining Bayesian state estimation and the encoding of uncertainty. It would be useful to have a figure dedicated to explaining this. In addition to the Paulin papers mentioned above, omissions include the work of Peter Latham (eg Ma et al. Bayesian inference with probabilistic population codes Nat. Neurosci. 2006.) and the work of Bence Ölveczky - e.g. Dhawale et al. The Role of Variability in Motor Learning Annu Rev Neurosci. 2017). In particular, given the proposed high dimensional codes in the granule cell population, uncertainty can be represented at the level of the granule cell population code; although this does not exclude the role of inhibition in shaping the code, its role may depend on how the statistics of mossy fibre inputs themselves change with context.
- 2) In section Precision on State estimation, it is discussed that the ability to change granule cell integrative properties can be modulate by inhibition, as a function of behavioural context (example of dampened auditory inputs in visuomotor task). It is not clear how this links to the precision-weighting, or whether this links to a more generalized framework of combining saliency and precision. As mentioned in part 1, a clearer description/example of how the framework of Bayesian state estimation should incorporate these contexts may be helpful.
- 3) Closed cortical-cerebellar loops are mentioned but the key paper on this was not cited (Kelly and Strick, J. Neurosci., 2003).
- 4) Predictive coding is mentioned but Kathleen Cullen, one of the main pioneers of this, is not cited - e.g. Brooks Carriot and Cullen, Nature Neurosci 2015.
- 5) Glutamate and GABA spillover within the cerebellar glomerulus is discussed but key original papers are omitted .
- 6) Given that the intrinsic activity of Golgi cells and the spatial properties of their inhibitory effect is discussed in some detail it is surprising that their strong local electrical coupling and its role in synchronizing and desynchronizing firing is not discussed. The strength, spatial dependence of gap junction coupling and synchrony were investigated in Dugue et al., Neuron 2009; Vervaeke et al., Neuron 2010; van Welie Neuron 2016).
- 7) Some studies that examined the behavioural outcome of eliminating Golgi cells, and altering granule cell inhibition would merit discussion. Example that come to mind include Seja et al. . EMBO J 2012; Chiu et al J. Neurosci. 2005 and Watanabe et al., Call 1998.
- 8) Is inhibition proposed to strictly increase with uncertainty? Its link to variability of inputs may be more stable than its link to behavioural or sensory uncertainty (depending on input statistics).
- 9) In the section "Golgi cells underlie precise granular layer computations", paragraphs 3-4, it is not clearly stated what is known physiological and anatomical evidence, versus what are the proposed or consensus implications about inhibitory control of the circuit. For example, the existence of potential strong feedback circuit is shown in [61], but the ability to dynamically modulate granule cells in this feedback manner (at the individual or network level) is still an untested hypothesis, especially given the variable kinetics at Golgi cell - granule cell synapses

[63]. While the downstream effects of inhibition on individual granule cells is well studied, the interaction of different motifs to shape inhibition accordingly needs further experimental corroboration.

10) It is unclear why the adaptive filter theory should be completely at odds with the precision-weighting. Instead of assuming a stable or uniform set of temporal basis functions, inhibition can change the dynamical repertoire of the network, that constrain or shape estimation at the output layer. Further, the experimental studies all report granule cell responses to external sensory stimuli, without any learning or predictive task component.

Decision letter (RSPB-2020-2290.R0)

19-Oct-2020

Dear Dr Palacios:

I am writing to inform you that your manuscript RSPB-2020-2290 entitled "Accounting for uncertainty: inhibition for neural inference in the cerebellum" has, in its current form, been rejected for publication in Proceedings B. The referees think the article could be useful and the topic is certainly interesting, but both have major concerns about the structure and the balance between novelty and being overly speculative. They also point to gaps in the literature cited,

So, it seems to me that substantial revisions are necessary and fair bit of work would be needed to reach the necessary standard. Because the topic is important, I would be willing to consider a resubmission, provided the comments of the referees are fully addressed. However please note that this is neither a provisional acceptance nor a task that can be done quickly and easily. You may decide that it's easier to submit elsewhere.

A resubmission would be treated as a new manuscript. However, we will approach the same reviewers if they are available and it is deemed appropriate to do so by the Editor. Please note that resubmissions must be submitted within six months of the date of this email. In exceptional circumstances, extensions may be possible if agreed with the Editorial Office. Manuscripts submitted after this date will be automatically rejected.

- 1) A 'response to referees' document including details of how you have responded to the comments, and the adjustments you have made.
- 2) A clean copy of the manuscript and one with 'tracked changes' indicating your 'response to referees' comments document.
- 3) Line numbers in your main document.
- 4) Please read our data sharing policies to ensure that you meet our requirements <https://royalsociety.org/journals/authors/author-guidelines/#data>.

Best wishes,
Innes Cuthill

Prof. Innes Cuthill
 Reviews Editor, Proceedings B
 mailto: proceedingsb@royalsociety.org

Reviewer(s)' Comments to Author:

Referee: 1

Comments to the Author(s)

This manuscript reviews helpful material related to the cerebellar Golgi cell - granule cell network and places that material in a probabilistic inference framework. As the authors note, in prevailing theory, the cerebellum is thought to execute forward model prediction. Although prediction requires incorporating uncertainty, the most common formulations of these ideas in cerebellar neuroscience often do not afford much role for uncertainty. Among many other places where uncertainty should propagate through the cerebellar circuitry, the authors here consider its role at the input layer in "weighting" granule cell representations of different information sources via activation patterns of the network of inhibitory Golgi cells.

The topics reviewed are likely to be of use to cerebellar and non-cerebellar researchers, and the probabilistic framework is helpful in a field whose most dominant theories are largely non-probabilistic in nature. The authors also helpfully describe unanswered questions and predictions that may be tested in future research.

I have two general suggestions for the authors' revision.

First, the review's model/hypothesis and language is sometimes too speculative or nonspecific, veering a bit far from "review" territory into theoretical hypothesis territory. To take an example, P10 "tonic inhibition sets an optimal signal-to-noise ratio, acting as a prior over average or expected signal intensity" (as far as I know this is hypothetical, and even as a theory, it's not exactly clear that this solves a necessary uncertainty problem. Aren't there specific uncertainties associated with different modalities?). More generally, the organization of the review sometimes takes the form of a list of known features of Golgi anatomy followed by a guess as to how those features might contribute to probabilistic inference. It would be helpful for the authors to lay out, somewhere near the outset, the set of uncertainties they think are relevant to inference. They can refer to this framework as they make their way through the review of Golgi functional anatomy, rather than springing on readers wholly new probabilistic concepts throughout the paper. Second, in most people's minds, the state prediction relevant to cerebellar processing is done after the input layer, e.g. in the Purkinje cells. This review is framed as addressing the issue of confidence, but it's focused on confidence in the input variables the cerebellum uses for state estimation. I agree that input variance is important to determining confidence in the resulting state estimation. Still it should be more explicitly described to naive readers that this is the first step in the process of inference, and is distinct from computing predictions and the confidence level of those predictions (input confidence must be propagated through to the final distribution of state estimates etc). The rest of these computations are beyond the scope here, but it will help to give readers a concise sense of the overall framework.

Other specific comments:

P3 par2 "In this framework, inference and learning are based on probabilistic models entertained by the brain, representing somatic and environmental variables or states, their dynamical interactions and causal link to sensory input."

"Causal link" is not a great descriptor. If the task is probabilistic inference, it is typically not relevant whether the connection is causal or correlative.

p.5 par1- "The inhibitory network in the granular layer is simpler than in cerebral cortical regions..."

The authors' ground their perspective in neocortical top-down inferential inhibitory processing, but to do so fairly they should consider addressing a few things more completely.

First, at the anatomical level there is not really a compelling comparison to be made between the extremely complicated intra- and inter-areal top-down feedback modulation of input to the neocortex, and the very limited recurrent circuitry of the cerebellum. This should give more pause than implied in the present text, and suggests the comparison should be at most a conceptual analog rather than a mechanistic one.

On the other hand, classical cerebellar theories also don't do justice to what "top-down" processing does exist in the cerebellum (by treating the cerebellum as a feedforward structure without recurrent excitation, in some cases almost as precondition of the computational theory).

As the authors note, DCN provides recurrent mossy fiber inputs, and this may indeed be a route for a more sophisticated form of feedback modulation. It might be helpful to stress that this potential mechanism for modulating inputs is a substantial divergence from classical feedforward adaptive filtering theory, and a probabilistic inference framework may provide a starting point for explaining this anatomical detail that is 'inconvenient' in other theories.

P6 par1 - In the discussion of Golgi cells' wide axonal arborization vs narrow dendritic column, I'm not sure I view this as evidence of a lateral inhibition mechanism, at least as presented, nor necessarily as a discrepancy? Granule cell axons are ~1 mm long--perhaps narrow Golgi dendritic columns are sufficient to sample from a granule cell population as widely distributed as that contacted by the diffuse Golgi axonal branches?

P9 Par 3 in the proof opens with a strange semi-repeat of par2

The organization of the last two sections "Adaptive filter model" and "Kalman filter model" is a bit unclear/jarring. Is the purpose of these sections to contrast them with an alternative model in which Golgi cells contribute to Bayesian inference? Is it to explain their shortcomings? Is it to explain that if Golgi cells convey uncertainty, that can be incorporated into these models in order to solve their shortcomings? In addition, it is unfortunate that this section leads directly into the discussion, without any concluding and transitional statements, as it will doubtless confuse readers less familiar with this literature by distracting from the central narrative of the review.

P3 par2 "In neural networks, precision weights presynaptic input..." is asserted as a known truth rather than as a hypothesis (assuming that is the intention? Otherwise needs citations).

A number of assorted typos/missing words/grammatical issues/partial sentences/run-on sentences, please take a look throughout.

Referee: 2

Comments to the Author(s)

This perspective/review advances the hypothesis that the inhibitory mechanisms present in the granule cell layer of the cerebellum could act as a sensorimotor state estimator that encodes uncertainty. The article begins with a rather brief introduction to motor control, probabilistic internal models and state estimation. None of these are sufficiently 'unpacked' for readers unfamiliar with these concepts to understand them. The central question of how the cerebellar cortex might compute state estimation is clearly stated and the proposal to examine the underlying mechanisms is set out. Moreover, the central idea of the cerebellar cortex encoding probability distributions that are used for Bayesian state estimation is interesting. But the authors seem to be unaware that it has been proposed before by Paulin (Paulin et al. 2005. *J. Neural Eng.* 2 (2005) S219-S234 doi:10.1088/1741-2560/2/3/S06), who has provided compelling arguments for state estimation since the early 1990s (Paulin *Human Movement Science* 12 (1993) 5-16; Paulin, *Brain Behav Evol* 1993;41:39-50). Overall, this manuscript provides a good review of the cellular and systems level studies of the cerebellar input layer (excepting those detailed below), but in my view it does not provide a convincing link between the underlying mechanisms and behaviours to the probabilistic state estimation hypothesis. The review thus provides a rather undeveloped description of encoding uncertainty and state estimation and a more complete account of the anatomy and physiology of inhibition the input layer and some descriptions of other theories of

cerebellar function. The figures have some artistic merit but convey little scientific content and miss the opportunity to explain the complex concepts discussed. Lastly the manuscript PDF has several major formatting errors. The main text was duplicated twice, with the references and conclusion missing from the initial copy of the text. Moreover, in the complete second copy there was a duplication of a large block of text in the section on the Kalman filter. In my view the manuscript requires a very major rewrite that includes an accessible explanation of exactly what probabilistic state estimation is, how uncertainty can be used in motor control and incorporates and acknowledges Paulin's and Latham's (see below) work on this. Moreover, the text needs to be much more focussed on the evidence and mechanisms that could support probabilistic state estimation as well as highlighting results that do not fit.

Specific points for consideration

- 1) Expand on explaining Bayesian state estimation and the encoding of uncertainty. It would be useful to have a figure dedicated to explaining this. In addition to the Paulin papers mentioned above, omissions include the work of Peter Latham (eg Ma et al. Bayesian inference with probabilistic population codes Nat. Neurosci. 2006.) and the work of Bence Ölveczky - e.g. Dhawale et al. The Role of Variability in Motor Learning Annu Rev Neurosci. 2017). In particular, given the proposed high dimensional codes in the granule cell population, uncertainty can be represented at the level of the granule cell population code; although this does not exclude the role of inhibition in shaping the code, its role may depend on how the statistics of mossy fibre inputs themselves change with context.
- 2) In section Precision on State estimation, it is discussed that the ability to change granule cell integrative properties can be modulate by inhibition, as a function of behavioural context (example of dampened auditory inputs in visuomotor task). It is not clear how this links to the precision-weighting, or whether this links to a more generalized framework of combining saliency and precision. As mentioned in part 1, a clearer description/example of how the framework of Bayesian state estimation should incorporate these contexts may be helpful.
- 3) Closed cortical-cerebellar loops are mentioned but the key paper on this was not cited (Kelly and Strick, J. Neurosci., 2003).
- 4) Predictive coding is mentioned but Kathleen Cullen, one of the main pioneers of this, is not cited - e.g. Brooks Carriot and Cullen, Nature Neurosci 2015.
- 5) Glutamate and GABA spillover within the cerebellar glomerulus is discussed but key original papers are omitted .
- 6) Given that the intrinsic activity of Golgi cells and the spatial properties of their inhibitory effect is discussed in some detail it is surprising that their strong local electrical coupling and its role in synchronizing and desynchronizing firing is not discussed. The strength, spatial dependence of gap junction coupling and synchrony were investigated in Dugue et al., Neuron 2009; Vervaeke et al., Neuron 2010; van Welie Neuron 2016).
- 7) Some studies that examined the behavioural outcome of eliminating Golgi cells, and altering granule cell inhibition would merit discussion. Example that come to mind include Seja et al. . EMBO J 2012; Chiu et al J. Neurosci. 2005 and Watanabe et al., Call 1998.
- 8) Is inhibition proposed to strictly increase with uncertainty? Its link to variability of inputs may be more stable than its link to behavioural or sensory uncertainty (depending on input statistics).
- 9) In the section "Golgi cells underlie precise granular layer computations", paragraphs 3-4, it is not clearly stated what is known physiological and anatomical evidence, versus what are the proposed or consensus implications about inhibitory control of the circuit. For example, the existence of potential strong feedback circuit is shown in [61], but the ability to dynamically

modulate granule cells in this feedback manner (at the individual or network level) is still an untested hypothesis, especially given the variable kinetics at Golgi cell – granule cell synapses [63]. While the downstream effects of inhibition on individual granule cells is well studied, the interaction of different motifs to shape inhibition accordingly needs further experimental corroboration.

10) It is unclear why the adaptive filter theory should be completely at odds with the precision-weighting. Instead of assuming a stable or uniform set of temporal basis functions, inhibition can change the dynamical repertoire of the network, that constrain or shape estimation at the output layer. Further, the experimental studies all report granule cell responses to external sensory stimuli, without any learning or predictive task component.

Author's Response to Decision Letter for (RSPB-2020-2290.R0)

See Appendix A.

RSPB-2021-0276.R0

Review form: Reviewer 2

Recommendation

Accept as is

Scientific importance: Is the manuscript an original and important contribution to its field?

Excellent

General interest: Is the paper of sufficient general interest?

Excellent

Quality of the paper: Is the overall quality of the paper suitable?

Excellent

Is the length of the paper justified?

Yes

Should the paper be seen by a specialist statistical reviewer?

No

Do you have any concerns about statistical analyses in this paper? If so, please specify them explicitly in your report.

No

It is a condition of publication that authors make their supporting data, code and materials available - either as supplementary material or hosted in an external repository. Please rate, if applicable, the supporting data on the following criteria.

Is it accessible?

N/A

Is it clear?

N/A

Is it adequate?

N/A

Do you have any ethical concerns with this paper?

No

Comments to the Author

The authors have done a good job in revising the article, which now acknowledges previous work on this subject and incorporates the latest research. There are however a few minor typos to correct.

Ln 144 F-I not defined.

Ln 165 '6' should be subscript

Ln 205 Huang et al., eLife 2013 could be added here.

Ln 327 HaMori. The 'M' should not be capitalized.

Decision letter (RSPB-2021-0276.R0)

24-Feb-2021

Dear Dr Palacios

I am pleased to inform you that your revised manuscript RSPB-2021-0276 entitled "Accounting for uncertainty: inhibition for neural inference in the cerebellum" has been accepted for publication in Proceedings B.

The referee is happy with your revisions and has recommended publication, but also pointed out some minor typos. Therefore, I invite you to make the changes and upload the final version of your manuscript. Because the schedule for publication is very tight, it is a condition of publication that you submit the revised version of your manuscript within 7 days. If you do not think you will be able to meet this date please let us know.

To upload your manuscript, log into <https://mc.manuscriptcentral.com/prsb> and enter your Author Centre, where you will find your manuscript title listed under "Manuscripts with Decisions." Under "Actions," click on "Create a Revision." Your manuscript number has been appended to denote a revision. You will be unable to make your revisions on the originally submitted version of the manuscript. Instead, revise your manuscript and upload a new version through your Author Centre.

When submitting your revised manuscript, you will be able to respond to the comments made by the referee and upload a file "Response to Referees". You can use this to confirm that you've corrected the typos.

1) A text file of the manuscript (doc, txt, rtf or tex), including the references, tables (including captions) and figure captions. Please remove any tracked changes from the text before submission. PDF files are not an accepted format for the "Main Document".

2) A separate electronic file of each figure (tiff, EPS or print-quality PDF preferred). The format should be produced directly from original creation package, or original software format. PowerPoint files are not accepted.

3) Electronic supplementary material: this should be contained in a separate file and where possible, all ESM should be combined into a single file. All supplementary materials accompanying an accepted article will be treated as in their final form. They will be published alongside the paper on the journal website and posted on the online figshare repository. Files on figshare will be made available approximately one week before the accompanying article so that the supplementary material can be attributed a unique DOI.

Best wishes,
Innes Cuthill

Prof. Innes Cuthill
Reviews Editor, Proceedings B
mailto: proceedingsb@royalsociety.org

Reviewer(s)' Comments to Author:
Referee: 2

Comments to the Author(s)

The authors have done a good job in revising the article, which now acknowledges previous work on this subject and incorporates the latest research. There are however a few minor typos to correct.

Ln 144 F-I not defined.
Ln 165 '6' should be subscript
Ln 205 Huang et al., eLife 2013 could be added here.
Ln 327 HaMori. The 'M' should not be capitalized.

Author's Response to Decision Letter for (RSPB-2021-0276.R0)

See Appendix B.

Decision letter (RSPB-2021-0276.R1)

01-Mar-2021

Dear Mr Palacios

I am pleased to inform you that your manuscript entitled "Accounting for uncertainty: inhibition for neural inference in the cerebellum" has been accepted for publication in Proceedings B.

If you are likely to be away from e-mail contact during this period, let us know. Due to rapid publication and an extremely tight schedule, if comments are not received, we may publish the paper as it stands.

Open access

You are invited to opt for open access via our author pays publishing model. Payment of open access fees will enable your article to be made freely available via the Royal Society website as soon as it is ready for publication. For more information about open access publishing please visit our website at http://royalsocietypublishing.org/site/authors/open_access.xhtml.

The open access fee is £1,700 per article (plus VAT for authors within the EU). If you wish to opt for open access then please let us know as soon as possible.

Paper charges

Sincerely,
Proceedings B
mailto:proceedingsb@royalsociety.org

Appendix A

Reply to referees

We would like to thank the referees for their insightful and useful comments, which greatly improved the review. We hope that the changes described below correspond to what the referees had in mind.

1st referee:

This manuscript reviews helpful material related to the cerebellar Golgi cell - granule cell network and places that material in a probabilistic inference framework. As the authors note, in prevailing theory, the cerebellum is thought to execute forward model prediction. Although prediction requires incorporating uncertainty, the most common formulations of these ideas in cerebellar neuroscience often do not afford much role for uncertainty. Among many other places where uncertainty should propagate through the cerebellar circuitry, the authors here consider its role at the input layer in weighting granule cell representations of different information sources via activation patterns of the network of inhibitory Golgi cells.

The topics reviewed are likely to be of use to cerebellar and non-cerebellar researchers, and the probabilistic framework is helpful in a field whose most dominant theories are largely non-probabilistic in nature. The authors also helpfully describe unanswered questions and predictions that may be tested in future research.

I have two general suggestions for the authors revision.

First, the reviews model/hypothesis and language is sometimes too speculative or nonspecific, veering a bit far from review territory into theoretical hypothesis territory. To take an example, P10 “tonic inhibition sets an optimal signal-to-noise ratio, acting as a prior over average or expected signal intensity” (as far as I know this is hypothetical, and even as a theory, its not exactly clear that this solves a necessary uncertainty problem. Aren’t there specific uncertainties associated with different modalities?). More generally, the organization of the review sometimes takes the form of a list of known features of Golgi anatomy followed by a guess as to how those features might contribute to probabilistic inference. It would be helpful for the authors to lay out, somewhere near the outset, the set of uncertainties they think are relevant to inference. They can refer to this framework as they make their way through the review of Golgi functional anatomy, rather than springing on readers wholly new probabilistic concepts throughout the paper.

Thanks for these suggestions. We have re-expressed some of our claims throughout the paper to make their hypothetical nature more evident. We have also reorganised the text to make the structure of the paper easier to follow. In particular, we are now clearly stating how different neural mechanisms associated to Golgi cell inhibition (summarised in Figure 3) relate to precision-weighting for state estimation at the

beginning and conclusion of the section “Golgi cells underlie precise granular layer computations”; in short, they inform neural dynamics about precision of mossy fibre input in different ways. We now say (lines 137-158, 239-243 in the manuscript with highlighted changes):

“The inhibitory network in the cerebellar granular layer is simpler than in cerebral cortical regions, lacking cellular diversity and extremely complicated intra- and inter-areal top-down feedback modulation (cf. [1, 2]); nonetheless, it is suited to effectively balance excitation in granule cells. Golgi cells act through both a hyperpolarising current that lowers granule cell resting potential – efficiently thresholding or gating MF input – and through an increase in membrane conductance or shunting inhibition – associated with faster membrane dynamics and ultimately a biased sensitivity towards synchronous presynaptic activity [3, 4]. As a result, Golgi cells can set the excitability or responsiveness of granule cells, approximated by the operative point (position and slope) of their F-I curve, controlling propagation of MF activity within the cerebellar circuit. From a neural inference perspective, this propagation should be conditional upon the precision of information transmitted, implying that Golgi cell inhibition is sensitive to signals that are most relevant for present belief updating. It is therefore necessary to identify which mechanisms may inform granule cell excitability via inhibition in this context-dependent manner. One distinction mentioned above is between bottom-up and top-down sources of conditioning; beyond this, various biophysical mechanisms might determine how Golgi cells operate. In this section, we highlight those mechanisms that may allow Golgi cell inhibition to perform precision-weighting of the input. First, we see how inhibition sets neural gain to match average levels of activity. Then, we consider time-varying inhibition and its modulation by temporal and spatial properties of the input; in doing so, we characterise the temporal unfolding of MF activity and its spatial organisation as a proxy for its intrinsic (bottom-up) precision. Finally, we address mechanisms, such as modulation of Golgi cells by neuromodulators or nucleocortical projections, that do not directly depend or arise from current MF input, yet control how these are transmitted by changing the endogenous state of the granular layer; we refer to these as top-down mechanisms signalling expected precision of the input.”

...

“In conclusion, there appear to be a variety of mechanisms that could inform the granular layer about precision of MF input, irrespectively of the extremely diversified nature of those input. These mechanisms condition granule cell excitation through Golgi cell inhibition, which constitutes the unique local feedback of the network. In this sense, Golgi cells emerge as a crucial hub for precise state estimation in the cerebellar cortex (Figure 2).”

In addition, as remarked in the comment, there could be specific uncertainties associated to the different types of mossy fibre input, which in turn could be encoded differently. The inhibitory mechanisms revised here are common to the granular layer, and thus might be able to deal with these different types of uncertainties, translating them into a common granule cell population code. We address these issues at the end of section “Precision in state estimation” (lines 93-106):

“The cerebellar cortex receives input via MFs from virtually every part of the brain. This input is rich, encompassing multiple sensory and motor modalities [5, 6, 7, 8, 9] as well as cognitive domains [10]. Moreover, its nature can be both predictive (e.g. anticipatory reward-related signals) and postdictive (e.g. sensory feedback) [11, 12, 13], encompassing the entire period of movement execution (see e.g. [12]). Consequently, at any given time a huge amount of information can potentially be transmitted to the granular layer via MFs. However, only a fraction of this information is likely to be relevant at any given moment in time; this fact intimates the necessity for the cerebellar cortex to select or prioritise some and not other sources of input, so that only information that is relevant in a particular behavioural context can affect state estimation. For example, while engaged in a visuomotor task, postsynaptic responses to MFs conveying confounding auditory signals might be dampened. **With respect to the cerebellar cortex, precision encoded in various extracerebellar regions must be translated and implemented in a common way within the granule cell population, in a manner which is instrumental for state estimation, that is, causing downstream layers to appropriately respond to encoded precision. Accordingly, inhibition in the input layer of the cerebellum appears capable of operating these fundamental operations.**”

Second, in most people’s minds, the state prediction relevant to cerebellar processing is done after the input layer, e.g. in the Purkinje cells. This review is framed as addressing the issue of confidence, but it’s focused on confidence in the input variables the cerebellum uses for state estimation. I agree that input variance is important to determining confidence in the resulting state estimation. Still it should be more explicitly described to naive readers that this is the first step in the process of inference, and is distinct from computing predictions and the confidence level of those predictions (input confidence must be propagated through to the final distribution of state estimates etc). The rest of these computations are beyond the scope here, but it will help to give readers a concise sense of the overall framework.

Thanks for the remark. We are now stating more clearly which stage of the hierarchical inferential processing in the cerebellum we are focussing on. We have made the following changes in the introduction (lines 37-42):

“These ideas are long-standing, but it remains unresolved how various components of the cerebellum could specifically contribute to state estimation. **In general, only activity and plasticity of Purkinje cells, the output of the cerebellar cortex, have been associated with this computation; however, inferential processes occur all the way through the hierarchy of an internal probabilistic model. Here we consider the first step in cerebellar cortical state estimation, by proposing a role for inhibition in the granular layer.** This network, comprising about half of the neurons in the mammalian nervous system, relays all extracerebellar input that is directed via mossy fibres (MFs) to Purkinje cells [14, 15, 16] (Figure 1). The granular layer is made up of excitatory granule cells and inhibitory Golgi cells, and interactions between these two neuronal populations determine network responses to external (MF) perturbations. Two aspects are key to understand neural dynamics within this network: firstly granule cells are numerous but individually receive only a small number of inputs (four excitatory and four inhibitory connections each on average [17, 18]); secondly Golgi cells are sparse relative to granule cells, but each neuron contacts hundreds to

thousands of granule cells through an extended axonal arborisation. Thus, Golgi cell inhibition is likely to have a big impact on individual granule cell activity and putative inferential processes in the cerebellum.”

Other specific comments:

P3 par2 “In this framework, inference and learning are based on probabilistic models entertained by the brain, representing somatic and environmental variables or states, their dynamical interactions and causal link to sensory input.”

“Causal link” is not a great descriptor. If the task is probabilistic inference, it is typically not relevant whether the connection is causal or correlative.

Thanks for the comment. We now say (lines 65-67):

“In this framework, inference and learning are based on probabilistic models entertained by the brain, representing somatic and environmental variables or states, their dynamical interactions and link to sensory input [19]”

p.5 par1- “The inhibitory network in the granular layer is simpler than in cerebral cortical regions”

The authors’ ground their perspective in neocortical top-down inferential inhibitory processing, but to do so fairly they should consider addressing a few things more completely.

First, at the anatomical level there is not really a compelling comparison to be made between the extremely complicated intra- and inter-areal top-down feedback modulation of input to the neocortex, and the very limited recurrent circuitry of the cerebellum. This should give more pause than implied in the present text, and suggests the comparison should be at most a conceptual analog rather than a mechanistic one.

On the other hand, classical cerebellar theories also don’t do justice to what “top-down” processing does exist in the cerebellum (by treating the cerebellum as a feedforward structure without recurrent excitation, in some cases almost as precondition of the computational theory). As the authors note, DCN provides recurrent mossy fiber inputs, and this may indeed be a route for a more sophisticated form of feedback modulation. It might be helpful to stress that this potential mechanism for modulating inputs is a substantial divergence from classical feedforward adaptive filtering theory, and a probabilistic inference framework may provide a starting point for explaining this anatomical detail that is “inconvenient” in other theories.

Thanks for the useful comments. We now stress the difference between cortical and cerebellar cortical inhibition in the section “Golgi cells underlie precise granular layer computations” (lines 137-138):

“The inhibitory network in the cerebellar granular layer is simpler than in cerebral cortical regions, **lacking cellular diversity and extremely complicated intra- and inter-areal top-down feed-**

back modulation (cf. [1, 2]); nonetheless, it is suited to effectively balance excitation in granule cells. Golgi cells act through both a hyperpolarising current that lowers granule cell resting potential – efficiently thresholding or gating MF input – and through an increase in membrane conductance or shunting inhibition – associated with faster membrane dynamics and ultimately a biased sensitivity towards synchronous presynaptic activity [3, 4]. As a result, Golgi cells can set the excitability or responsiveness of granule cells, approximated by the operative point (position and slope) of their F-I curve, controlling propagation of MF activity within the cerebellar circuit. From a neural inference perspective, this propagation should be conditional upon the precision of information transmitted, implying that Golgi cell inhibition is **sensitive to signals that are most relevant for present belief updating**. **It is therefore necessary to identify which mechanisms may inform granule cell excitability via inhibition in this context-dependent manner. One distinction mentioned above is between bottom-up and top-down sources of conditioning; beyond this, various biophysical mechanisms might determine how Golgi cells operate. In this section, we highlight those mechanisms that may allow Golgi cell inhibition to perform precision-weighting of the input. First, we see how inhibition sets neural gain to match average levels of activity. Then, we consider time-varying inhibition and its modulation by temporal and spatial properties of the input; in doing so, we characterise the temporal unfolding of MF activity and its spatial organisation as a proxy for its intrinsic (bottom-up) precision. Finally, we address mechanisms, such as modulation of Golgi cells by neuromodulators or nucleocortical projections, that do not directly depend or arise from current MF input, yet control how these are transmitted by changing the endogenous state of the granular layer; we refer to these as top-down mechanisms signalling expected precision of the input.**”

And in the discussion (lines 261-269):

“Arguably, neural dynamics underlying state estimation in the cerebellar cortex should necessarily be sensitive to uncertainty associated with inference, as discussed more generally in the context of cortical functioning [20, 21, 22]. **Cortical control of excitation is much more complex than in the granular layer, hinting at a more sophisticated neural inference. Nonetheless, fine-tuning of granule cell activity via Golgi cells also seems to be calibrated by a variety of mechanisms, including top-down signals from cerebellar nuclei, as well as from downstream layers in the cerebellar cortex itself. This in turn should bear on any theory aiming to explain cerebellar computations. In particular, recurrent connectivity within it has usually been neglected or oversimplified; in contrast, a probabilistic inference framework may provide a starting point for explaining this anatomical detail, as shown in this review.**”

P6 par1 - In the discussion of Golgi cells’ wide axonal arborization vs narrow dendritic column, I’m not sure I view this as evidence of a lateral inhibition mechanism, at least as presented, nor necessarily as a discrepancy? Granule cell axons are 1 mm long—perhaps narrow Golgi dendritic columns are sufficient to sample from a granule cell population as widely distributed as that contacted by the diffuse Golgi axonal branches?

Thanks for highlighting this important point. We now explain that even if extention

from parallel fibre could confound lateral inhibition mechanisms, there is still evidence that local information from mossy fibres and granule cells (via ascending axons) could preferentially drive Golgi cells, supporting lateral inhibition via the mismatch between axon and dendrites. Besides, even assuming equal strength/kinetics between parallel fibres and ascending axons/mossy fibre synapses onto Golgi cells, the existence of different circuits supports the idea of a different use of local vs global information by Golgi cells (lines 192-222).

“Along with temporal features of Golgi cell inhibition, the spatial arrangement of Golgi cell processes may also play a role in the contextualisation of incoming information [23]. Notably, there is a mismatch between the narrow granular layer region from which Golgi cells receive excitatory inputs (determined by the dendritic tree), and the region extending hundreds of micrometers over which they exert inhibitory influence (determined by the axonal plexus). **In the present discussion, lateral inhibition could be linked to representational precision via its effects over correlations among different streams of MF input. Excitation-inhibition balance at any location in the granular layer could then reflect – via horizontal mixing of Golgi cell signals – the precision of the local information, relative to its surround.** In practice, this could lead to an increase of fast correlations among clusters of granule cells that are excited by common MFs, and a simultaneous decrease of slower correlations across **competing** patches of granular layer – replicating observations in structures that share a similar geometry, like the olfactory bulb [24].

This contextual modulation of granule cell excitability relies on spatial constraints of information driving Golgi and granule cell populations, which in turn depend on different anatomical properties of the network. MFs show substantial anisotropic divergence in the granular layer [25], which enables integration of various sources of information at the level of single granule cells, but prevents the emergence of ordered, neocortical-like receptive fields. As a consequence, fast correlations among Golgi cells (and inhibited clusters of granule cells) sharing MF input might be more evident within distributed, scattered groups of cells [26].

Another important anatomical property is the presence of millimeter-long granule-Golgi cell connections mediated by parallel fibres [27]. Parallel fibres have been linked to extended oscillations in the granular layer during rest [28], possibly setting a global pace for network computations and dynamics. Notably, these connections appear to be qualitatively different from local contacts made by ascending granule cell axons onto Golgi cells, which resemble more the faster and stronger MF-Golgi cell synapses [29, 30]. It follows that upon localised activation of MF terminals, parallel fibres might preferentially contribute to slow correlation of granule cells across the transverse axis [31], while ascending axons precisely entrain spiking of surrounding Golgi cells. Furthermore, the existence of electrical connections among Golgi cells further increases their sensitivity to temporal coincidence of local excitation, enhancing synchrony or alternatively asynchrony in and between granule cell clusters [32, 33, 34]. Therefore, different degrees of correlations might coexist in the granular layer, following properties of MF input and connectivity structure within the network, which might result in balanced dynamics of excitation and inhibition reflecting the statistics (precision) of information encoded.”

P9 Par 3 in the proof opens with a strange semi-repeat of par2

Thanks for noticing this. This paragraph is no longer present.

The organization of the last two sections Adaptive filter model and Kalman filter model is a bit unclear/jarring. Is the purpose of these sections to contrast them with an alternative model in which Golgi cells contribute to Bayesian inference? Is it to explain their shortcomings? Is it to explain that if Golgi cells convey uncertainty, that can be incorporated into these models in order to solve their shortcomings? In addition, it is unfortunate that this section leads directly into the discussion, without any concluding and transitional statements, as it will doubtless confuse readers less familiar with this literature by distracting from the central narrative of the review.

Thanks for calling our attention to these problems. Unfortunately, because of limited word count, we had to omit this section.

P3 par2 In neural networks, precision weights presynaptic input... is asserted as a known truth rather than as a hypothesis (assuming that is the intention? Otherwise needs citations).

Thanks for signalling this. We have added the citation and clarified how this follows from a view of neural dynamics as supporting inference (lines 64-79).

“Many aspects of brain functioning can be phrased in terms of probabilistic inference and learning processes [35, 36, 37]. In this framework, inference and learning are based on probabilistic models entertained by the brain, representing somatic and environmental variables or states, their dynamical interactions and link to sensory input [19]. Central to this argument is the notion of uncertainty, describing the spread or variance of belief distributions **assumed to be implicitly encoded by neural activity. Whatever the exact form of this encoding, one can argue that the variance of the implicit distributions depends in first place on the quality of data available to the network. In other words, uncertainty represented in neural activity should be a function of input precision, a measure of the reliability of input that determines how much this drives belief updating (Box).**

In biological neural networks, precision naturally translates into population gain [21], which scales or weights presynaptic input and adjusts its capacity to elicit voltage changes in the target **population.** The underlying idea is that a neural circuit is a system with endogenous or autonomous dynamics, whose activity is not entirely determined by external stimuli; its response to events can contextually vary, conditioned on their precision. **Here we assume that inputs reporting more precise representations are associated with higher population gain, that is, a stronger impact on downstream network dynamics – whose output in turn is implicitly linked to more precise distributions.”**

A number of assorted typos/missing words/grammatical issues/partial sentences/run-on sentences, please take a look throughout.

We have amended these errors.

2nd referee:

This perspective/review advances the hypothesis that the inhibitory mechanisms present in the granule cell layer of the cerebellum could act as a sensorimotor state estimator that encodes uncertainty. The article begins with a rather brief introduction to motor control, probabilistic internal models and state estimation. None of these are sufficiently unpacked for readers unfamiliar with these concepts to understand them.

Thanks for highlighting this: we have split the first paragraph of the introduction to better express the link between sensorimotor control, internal models for state estimation and uncertainty in those estimates (lines 22-36 in the manuscript with highlighted changes):

“Sensorimotor coordination or control can be regarded as the realisation of expected sensation via movement. It involves interactions between an agent and its environment; like when a mouse is actively gathering information with its whiskers. In order to control these interactions, the brain must be able to approximate or predict the consequences of forthcoming action. This relies on accurate estimates of behaviourally relevant states (such as whisker position) generated by an underlying model of how states relate to one another. Estimates are intrinsically uncertain, reflecting stochasticity in sensory channels and dynamics of states. Hence, when considering the neural implementation of an estimation process, it is desirable that neural circuits are capable of representing estimates conditioned on their associated uncertainty; in other words, the underlying models ought to be probabilistic.

The cerebellum has long been posited to instantiate probabilistic internal models for estimation of rapidly varying external states [38], whether somatic, such as limb kinematics [39], or environmental, for example moving targets [40]. In the cerebellum, these models are deemed to support sensorimotor control [41, 42, 43], as well as more abstract mental representations [44], by complementing ongoing neural computations in other brain regions with internally generated, delay-free probabilistic estimates of stochastic external dynamics, built upon past experience and integrating multiple sources of noisy information.”

The central question of how the cerebellar cortex might compute state estimation is clearly stated and the proposal to examine the underlying mechanisms is set out. Moreover, the central idea of the cerebellar cortex encoding probability distributions that are used for Bayesian state estimation is interesting. But the authors seem to be unaware that it has been proposed before by

Paulin (Paulin et al. 2005. J. Neural Eng. 2 (2005) S219S234 doi:10.1088/1741-2560/2/3/S06), who has provided compelling arguments for state estimation since the early 1990s (Paulin Human Movement Science 12 (1993) 5-16; Paulin, Brain Behav Evol 1993;41:39-50).

Overall, this manuscript provides a good review of the cellular and systems level studies of the cerebellar input layer (excepting those detailed below), but in my view it does not provide a convincing link between the underlying mechanisms and behaviours to the probabilistic state estimation hypothesis. The review thus provides a rather undeveloped description of encoding uncertainty and state estimation and a more complete account of the anatomy and physiology of inhibition the input layer and some descriptions of other theories of cerebellar function. The figures have some artistic merit but convey little scientific content and miss the opportunity to explain the complex concepts discussed.

Lastly the manuscript PDF has several major formatting errors. The main text was duplicated twice, with the references and conclusion missing from the initial copy of the text. Moreover, in the complete second copy there was a duplication of a large block of text in the section on the Kalman filter. In my view the manuscript requires a very major rewrite that includes an accessible explanation of exactly what probabilistic state estimation is, how uncertainty can be used in motor control and incorporates and acknowledges Paulin’s and Latham’s (see below) work on this. Moreover, the text needs to be much more focussed on the evidence and mechanisms that could support probabilistic state estimation as well as highlighting results that do not fit.

Thanks for the detailed comments. We have now expanded the description of uncertainty encoding and state estimation and made the figures more pertinent (see replies to specific points of consideration below). Now we also acknowledge papers from Paulin in the introduction (line 32):

“The cerebellum has long been posited to instantiate probabilistic internal models for estimation of rapidly varying external states [38], whether somatic, such as limb kinematics [39], or environmental, for example moving targets [40]. In the cerebellum, these models are deemed to support sensorimotor control [41, 42, 43], as well as more abstract mental representations [44], by complementing ongoing neural computations in other brain regions with internally generated, delay-free probabilistic estimates of stochastic external dynamics, built upon past experience and integrating multiple sources of noisy information.”

And in the box (128):

“In order for network dynamics in the cerebellum to reflect inferential processes, it is necessary that uncertainty in state estimation influences neural activity. There exist different models of how probability distributions can be encoded by neural populations (see [45] for an example in the cerebellum), some of which highlight the possibility that neural activity scales with the precision of the encoded distribution [46], while becoming sparser as an effect of divisive normalisation [47] (panel **B**). Notably, activity levels in recipient populations result from a combination of input and population gain/responsiveness, which here we associate with input precision. In other words, we argue that (un)certainty in neural representations is determined by the quality of information in the input driving those representations (panel **C**). Notice that input precision-weighting relies on the capacity of the network to assess this precision. Here we address this possibility and propose

that Golgi cells in the cerebellar granular layer mediate the link between neural dynamics and state estimation, by making network excitability sensitive to and reflective of uncertainty in inferential processes.”

Specific points for consideration

1) Expand on explaining Bayesian state estimation and the encoding of uncertainty. It would be useful to have a figure dedicated to explaining this. In addition to the Paulin papers mentioned above, omissions include the work of Peter Latham (eg Ma et al. Bayesian inference with probabilistic population codes Nat. Neurosci. 2006.) and the work of Bence Iveczky - e.g. Dhawale et al. The Role of Variability in Motor Learning Annu Rev Neurosci. 2017). In particular, given the proposed high dimensional codes in the granule cell population, uncertainty can be represented at the level of the granule cell population code; although this does not exclude the role of inhibition in shaping the code, its role may depend on how the statistics of mossy fibre inputs themselves change with context.

Thanks for indicating this shortcoming in the review. We expanded the explanation of Bayesian state estimation and uncertainty encoding with text in the box and changed the figure to make this information more explicit (lines 109-133). We also cite relevant references (line 126)

Box1 “When investigating the functions of a neural network, we usually try to identify which features of the body or world are encoded in the activity of its constituent neurons. Underlying this approach is the assumption that there is a mapping between the activity of the network and the outer states. Because this mapping is indirect – mediated by vicarious input about the system – it licenses an interpretation of neural circuits as internal models inferring causes of their input, like a patch of V1 reflecting the possible presence of a luminous bar projected onto the visual field. Importantly, this mapping is necessarily probabilistic, because the dynamics and interactions between states and sensory signals are noisy. By accounting for this stochasticity, neuronal activity comes to reflect probability distributions over states.

The cerebellum is thought to instantiate internal models for motor and cognitive calibration and adaptation. Neural activity in this region has indeed been observed to accurately encode dynamics of somatic or environmental states, such as whisker position in the mouse cerebellum [48]. These representations in turn contribute to sensorimotor coordination by refining motion [49] and sustaining or altering neural activity in other brain regions, such as the neocortex [50, 51, 52, 53] (panel **A**).

In order for network dynamics in the cerebellum to reflect inferential processes, it is necessary that uncertainty in state estimation influences neural activity. There exist different models of how probability distributions can be encoded by neural populations (see [45] for an example in the cerebellum), some of which highlight the possibility that neural activity scales with the precision of the encoded distribution [46], while becoming sparser as an effect of divisive normalisation [47] (panel **B**). Notably, activity levels in recipient populations result from a combination of input and population gain/responsiveness, which here we associate with input precision. In other words, we

argue that (un)certainty in neural representations is determined by the quality of information in the input driving those representations (panel **C**). Notice that input precision-weighting relies on the capacity of the network to assess this precision. Here we address this possibility and propose that Golgi cells in the cerebellar granular layer mediate the link between neural dynamics and state estimation, by making network excitability sensitive to and reflective of uncertainty in inferential processes.”

2) In section Precision on State estimation, it is discussed that the ability to change granule cell integrative properties can be modulate by inhibition, as a function of behavioural context (example of dampened auditory inputs in visuomotor task). It is not clear how this links to the precision-weighting, or whether this links to a more generalized framework of combining saliency and precision. As mentioned in part 1, a clearer description/example of how the framework of Bayesian state estimation should incorporate these contexts may be helpful.

Along with changes in the box (reply to 1)), we have made the following changes to the section “Precision in state estimation” in order to make clearer the relationship between gain modulation in neural network activity and precision-weighting in state estimation, and how this connects to behavioural contingences (lines 64-106).

“Many aspects of brain functioning can be phrased in terms of probabilistic inference and learning processes [35, 36, 37]. In this framework, inference and learning are based on probabilistic models entertained by the brain, representing somatic and environmental variables or states, their dynamical interactions and link to sensory input [19]. Central to this argument is the notion of uncertainty, describing the spread or variance of belief distributions **assumed to be implicitly encoded by neural activity**. Whatever the exact form of this encoding, one can argue that the variance of the implicit distributions depends in first place on the quality of data available to the network. In other words, uncertainty represented in neural activity should be a function of input precision, a measure of the reliability of input that determines how much this drives belief updating (Box).

In biological neural networks, precision naturally translates into population gain [21], which scales or weights presynaptic input and adjusts its capacity to elicit voltage changes in the target population. The underlying idea is that a neural circuit is a system with endogenous or autonomous dynamics, whose activity is not entirely determined by external stimuli; its response to events can contextually vary, conditioned on their precision. **Here we assume that inputs reporting more precise representations are associated with higher population gain, that is, a stronger impact on downstream network dynamics – whose output in turn is implicitly linked to more precise distributions.**

This brings us to two key points: first, the quality or precision of information is not reducible to its content, meaning that neural mechanisms signalling *what* is represented can be different from those signalling *how* it should be represented. For instance, the identity and activity pattern of upstream neurons can be related to the nature of a stimulus encoded, whereas the postsynaptic gain to the amount of information transmitted. Second, the precision of an input, realised as population gain and ultimately translated into patterns of excitation-inhibition balance, is related to the concurrent behavioural context. This can be exemplified through attentional gain modulation in

visual and auditory cortex [20, 21, 54], where contextualisation (**weighting**) of sensory stimuli by their precision can be accomplished via temporal coincidence of pre and postsynaptic activity, increasing probability of conversion of pre to postsynaptic spikes. In this case, **gain-by-synchrony** depends on both bottom-up (e.g. intrinsic saliency of the stimulus) and top-down (attentional) effects [55, 56, 57], **which are a function of behaviour. It follows that the extent to which a stimulus can be relevant for ongoing inference – under a certain behaviour – translates into precision-weighting of that stimulus via gain modulation.**

The cerebellar cortex receives input via MFs from virtually every part of the brain. This input is rich, encompassing multiple sensory and motor modalities [5, 6, 7, 8, 9] as well as cognitive domains [10]. Moreover, its nature can be both predictive (e.g. anticipatory reward-related signals) and postdictive (e.g. sensory feedback) [11, 12, 13], encompassing the entire period of movement execution (see e.g. [12]). Consequently, at any given time a huge amount of information can potentially be transmitted to the granular layer via MFs. However, only a fraction of this information is likely to be relevant at any given moment in time; this fact intimates the necessity for the cerebellar cortex to select or prioritise some and not other sources of input, so that only information that is relevant in a particular behavioural context can affect state estimation. For example, while engaged in a visuomotor task, postsynaptic responses to MFs conveying confounding auditory signals might be dampened. **With respect to the cerebellar cortex, precision encoded in various extracerebellar regions must be translated and implemented in a common way within the granule cell population, in a manner which is instrumental for state estimation, that is, causing downstream layers to appropriately respond to encoded precision. Accordingly, inhibition in the input layer of the cerebellum appears capable of operating these fundamental operations.”**

3) Closed cortical-cerebellar loops are mentioned but the key paper on this was not cited (Kelly and Strick, J. Neurosci., 2003).

We have added the original paper on corcito-cerebellar loops (line 121).

“The cerebellum is thought to instantiate internal models for motor and cognitive calibration and adaptation. Neural activity in this region has indeed been observed to accurately encode dynamics of somatic or environmental states, such as whisker position in the mouse cerebellum [48]. These representations in turn contribute to sensorimotor coordination by refining motion [49] and sustaining or altering neural activity in other brain regions, such as the neocortex [50, 51, 52, 53] (panel **A**).”

4) Predictive coding is mentioned but Kathleen Cullen, one of the main pioneers of this, is not cited e.g. Brooks Carriot and Cullen, Nature Neurosci 2015.

Thanks for indicating the work of Kathleen Cullen. We have added this citation when introducing cerebellar-based internal probabilistic models (line 34):

“The cerebellum has long been posited to instantiate probabilistic internal models for estimation of rapidly varying external states [38], whether somatic, such as limb kinematics [39], or environmental, for example moving targets [40]. In the cerebellum, these models are deemed to support sensorimotor control [41, 42, 43], as well as more abstract mental representations [44], by complementing ongoing neural computations in other brain regions with internally generated, delay-free probabilistic estimates of stochastic external dynamics, built upon past experience and integrating multiple sources of noisy information.”

5) Glutamate and GABA spillover within the cerebellar glomerulus is discussed but key original papers are omitted.

Thanks for the remark; we now cite Rossi, Hamman, neuron 1998 for GABA spillover, DiGregorio, Nusser, Silver, Neuron 2003 and Nielsen, DiGregorio, Silver, Neuron 2004 for glutamate spillover (line 164).

“A substantial component of inhibition is tonic, hinging on constantly activated extrasynaptic receptors that are responsive to ambient levels of neurotransmitter concentration [58]. This persistent form of inhibition, arising in part from non-vesicular sources of GABA [59, 60], is favoured by the synaptic organisation of the granular layer. Most if not all synaptic connections to granule cells are indeed located in special structures called glomeruli, which form isolated microenvironments where neurotransmitter (both GABA and glutamate) can accumulate and easily diffuse [18, 61, 62, 63]. In these compartments, ambient concentrations of GABA are sufficient to persistently activate high-affinity $\alpha 6\delta$ -subunit containing GABAA receptors [64]. *In vivo*, tonic inhibition minimises granule cell responsiveness to uncorrelated, temporally scattered inputs [65], while maintaining an exquisite sensitivity to salient (e.g. sensory-evoked) stimuli [66]. Therefore, tonic inhibition appropriately fixes granule cell excitability to match the average levels of MF activity, establishing a slowly changing threshold on neural gain discriminating noise from signals. In mathematical terms, this may be equivalent to a prior over expected precision of the input required for its propagation. At a behavioural level, loss of motor coordination resulting from the disruption of tonic inhibition, for example due to alcohol consumption [67], might then reflect global alterations in representational uncertainty.”

6) Given that the intrinsic activity of Golgi cells and the spatial properties of their inhibitory effect is discussed in some detail it is surprising that their strong local electrical coupling and its role in synchronizing and desynchronizing firing is not discussed. The strength, spatial dependence of gap junction coupling and synchrony were investigated in Dugue et al., Neuron 2009; Vervaeke et al., Neuron 2010; van Welie Neuron 2016).

Thanks for highlighting this gap in the review. We now mention Golgi-Golgi cell electrical connections in the context of short- and long-range correlations and spatial localisation of information (lines 210-222).

“Another important anatomical property is the presence of millimeter-long granule-Golgi cell connections mediated by parallel fibres [27]. Parallel fibres have been linked to extended oscillations in the granular layer during rest [28], possibly setting a global pace for network computations and dynamics. Notably, these connections appear to be qualitatively different from local contacts made by ascending granule cell axons onto Golgi cells, which resemble more the faster and stronger MF-Golgi cell synapses [29, 30]. It follows that upon localised activation of MF terminals, parallel fibres might preferentially contribute to slow correlation of granule cells across the transverse axis [31], while ascending axons precisely entrain spiking of surrounding Golgi cells. Furthermore, the existence of electrical connections among Golgi cells further increases their sensitivity to temporal coincidence of local excitation, enhancing synchrony or alternatively asynchrony in and between granule cell clusters [32, 33, 34]. Therefore, different degrees of correlations might coexist in the granular layer, following properties of MF input and connectivity structure within the network, which might result in balanced dynamics of excitation and inhibition reflecting the statistics (precision) of information encoded.”

7) Some studies that examined the behavioural outcome of eliminating Golgi cells, and altering granule cell inhibition would merit discussion. Example that come to mind include Seja et al. . EMBO J 2012; Chiu et al J. Neurosci. 2005 and Watanabe et al., Call 1998.

Thanks for the remark and for reporting these relevant works. We now say in the last paragraph of the discussion (lines 270-285):

“Testing these ideas requires both a theoretical and experimental efforts. Here we have assumed a general principle, namely, that precision in neural representations should affect their propagation across the different stages of inference – by tuning population gain. However, future work should aim at investigate the exact nature of this probabilistic encoding throughout the cerebellar circuit. From the experimental side, testing these ideas requires tracking and manipulation of spatiotemporal properties of excitation-inhibition balance in the granular layer, as MF input is transformed into parallel fibre output. The exact shape of ensuing activity patterns depends on many factors, including kinetics variability at the Golgi-granule cell synapses [68], which might result from plastic mechanisms in the granular layer [69]. Previous works have examined the consequences of altering excitation levels in this network, showing a direct link to motor impairment, including tremors, ataxia and reduced reflex adaptation [67, 70, 71]. Interestingly, Golgi cell ablation alters the spatiotemporal patterns of activity in the granule cell population without necessarily producing overexcitation, as the result of compensatory mechanisms such as reduced NMDA activity [72]. This highlights the importance of fine-grained granule cell activity patterns for downstream computations [71] – here argued to be the result of Golgi cell-mediated precision-weighting at the first stage of state estimation in the cerebellar cortex. Ultimately, technical advances in the field [73, 74] will make possible to verify or not these ideas.”

8) Is inhibition proposed to strictly increase with uncertainty? Its link to variability of inputs may

be more stable than its link to behavioural or sensory uncertainty (depending on input statistics).

We describe a possible link between uncertainty in mossy fibre input and granule cell population gain mediated by Golgi cell inhibition. This implies that on average excitatory and inhibitory currents in granule cells might become more balanced with higher uncertainty, and more skewed with lower uncertainty. Also, this reasoning requires a link between behavioural and sensory uncertainty and the uncertainty implicit in mossy fibre input, which can be encoded in its statistics (e.g. synchrony and frequency). We address these points in the section “Precision in state estimation” (lines 64-92)

“Many aspects of brain functioning can be phrased in terms of probabilistic inference and learning processes [35, 36, 37]. In this framework, inference and learning are based on probabilistic models entertained by the brain, representing somatic and environmental variables or states, their dynamical interactions and link to sensory input [19]. Central to this argument is the notion of uncertainty, describing the spread or variance of belief distributions **assumed to be implicitly encoded by neural activity**. Whatever the exact form of this encoding, one can argue that the variance of the implicit distributions depends in first place on the quality of data available to the network. In other words, uncertainty represented in neural activity should be a function of input precision, a measure of the reliability of input that determines how much this drives belief updating (Box).

In biological neural networks, precision naturally translates into population gain [21], which scales or weights presynaptic input and adjusts its capacity to elicit voltage changes in the target population. The underlying idea is that a neural circuit is a system with endogenous or autonomous dynamics, whose activity is not entirely determined by external stimuli; its response to events can contextually vary, conditioned on their precision. **Here we assume that inputs reporting more precise representations are associated with higher population gain, that is, a stronger impact on downstream network dynamics – whose output in turn is implicitly linked to more precise distributions.**

This brings us to two key points: first, the quality or precision of information is not reducible to its content, meaning that neural mechanisms signalling *what* is represented can be different from those signalling *how* it should be represented. For instance, the identity and activity pattern of upstream neurons can be related to the nature of a stimulus encoded, whereas the postsynaptic gain to the amount of information transmitted. Second, the precision of an input, realised as population gain and ultimately translated into patterns of excitation-inhibition balance, is related to the concurrent behavioural context. This can be exemplified through attentional gain modulation in visual and auditory cortex [20, 21, 54], where contextualisation (**weighting**) of sensory stimuli by their precision can be accomplished via temporal coincidence of pre and postsynaptic activity, increasing probability of conversion of pre to postsynaptic spikes. In this case, **gain-by-synchrony** depends on both bottom-up (e.g. intrinsic saliency of the stimulus) and top-down (attentional) effects [55, 56, 57], **which are a function of behaviour. It follows that the extent to which a stimulus can be relevant for ongoing inference – under a certain behaviour – translates into precision-weighting of that stimulus via gain modulation.”**

And in the caption of figure 3:

“Tuning of the system’s excitability controls precision of representations. Left: the excitation-inhibition balance in the granular layer (circular blue and pink arrows) is a function of both MF input and neural mechanisms signaling its precision by tuning Golgi cell inhibition. Right: within a population, specific neurons might exhibit higher or lower synaptic gain, depending for instance on stimulus overlap with their receptive field, while at a network level, population gain associated with precision of upstream representations dictates the responsiveness of neural ensembles. Golgi cell inhibition sets population gain, such that the balance of excitation and inhibition in granule cells reflects precision-weighted input; and encodes neural representations whose precision determines their transmission and influence on downstream integrative layers via parallel fibres. In **A**, MF input is coupled with high population gain and strongly drives granule cells, pushing excitation (pink bar) to overcome inhibition (blue bar). The ensuing population activity then represents states with high precision, exemplified by the red distribution. In **B**, MF input convey less reliable information, and the low gain brings inhibition to balance excitation, making the network almost unresponsive. The small network output, in turn, encodes state estimates with low precision, which will not be effective in driving neural inference downstream.”

9) In the section Golgi cells underlie precise granular layer computations, paragraphs 3-4, it is not clearly stated what is known physiological and anatomical evidence, versus what are the proposed or consensus implications about inhibitory control of the circuit. For example, the existence of potential strong feedback circuit is shown in [61], but the ability to dynamically modulate granule cells in this feedback manner (at the individual or network level) is still an untested hypothesis, especially given the variable kinetics at Golgi cell granule cell synapses [63]. While the downstream effects of inhibition on individual granule cells is well studied, the interaction of different motifs to shape inhibition accordingly needs further experimental corroboration.

Thanks for the remark. We have re-expressed some of our claims throughout this section to make their hypothetical nature more evident (lines 159-243).

“A substantial component of inhibition is tonic, hinging on constantly activated extrasynaptic receptors that are responsive to ambient levels of neurotransmitter concentration [58]. This persistent form of inhibition, arising in part from non-vesicular sources of GABA [59, 60], is favoured by the synaptic organisation of the granular layer. Most if not all synaptic connections to granule cells are indeed located in special structures called glomeruli, which form isolated microenvironments where neurotransmitter (both GABA and glutamate) can accumulate and easily diffuse [18, 61, 62, 63]. In these compartments, ambient concentrations of GABA are sufficient to persistently activate high-affinity $\alpha 6\delta$ -subunit containing GABAA receptors [64]. *In vivo*, tonic inhibition minimises granule cell responsiveness to uncorrelated, temporally scattered inputs [65], while maintaining an exquisite sensitivity to salient (e.g. sensory-evoked) stimuli [66]. Therefore, tonic inhibition appropriately fixes granule cell excitability to match the average levels of MF activity, establishing a slowly changing threshold on neural gain discriminating noise from signals. In mathematical terms, this may be equivalent to a prior over expected precision of the input required for its propagation. At a behavioural level, loss of motor coordination resulting from the disruption of tonic inhibition, for example due to alcohol consumption [67], might then reflect global alterations in representational uncertainty.

On top of a persistent inhibitory conductance, feedforward and feedback synaptic loops enable Golgi cells to dynamically modulate granule cells by following rapid variations in network activity [30] – although the exact contribution of these loops is still unknown. Phasic inhibition underlies balanced dynamics of excitation and inhibition in granule cells. Notably, phasic inhibition from Golgi cells can promptly track changes in MF spiking behaviour while, at the same time, accumulate over Golgi cell spike trains to match input firing rates [63, 68, 75, 23]. The ensuing coordination of excitation and inhibition, on a timescale ranging from few to hundreds of milliseconds, can determine which input patterns elicit responses based on the evoked instantaneous balance. Accordingly, when inhibition is temporally matched to excitation, granule cell firing is reduced but becomes more similar across cells [76]: *in vivo*, this could favour for example selective transmission of the synchronous and invariant component of MF stimuli to Purkinje cells, by virtue of its stronger impact on postsynaptic neurons. Moreover, inhibition can preserve temporal information in granule cell output by rapidly trailing excitation and forcing a sharp integration window of couple of milliseconds for EPSCs [75]. Overall, balanced dynamics could increase the capacity of granule cells to reliably transmit temporally structured information – here associated with high precision representations. This is in agreement with the general observation that the granular layer faithfully encodes extracerebellar activity [77, 78, 79, 80, 81]; and resonates with the idea of a precision-weighting mechanism relying on inhibition and sensitive to bottom-up dynamics, such as synchrony in MF input enhancing temporal coordination across subsets of Golgi cells [34].

Along with temporal features of Golgi cell inhibition, the spatial arrangement of Golgi cell processes may also play a role in the contextualisation of incoming information [23]. Notably, there is a mismatch between the narrow granular layer region from which Golgi cells receive excitatory inputs (determined by the dendritic tree), and the region extending hundreds of micrometers over which they exert inhibitory influence (determined by the axonal plexus). In the present discussion, lateral inhibition could be linked to representational precision via its effects over correlations among different streams of MF input. Excitation-inhibition balance at any location in the granular layer could then reflect – via horizontal mixing of Golgi cell signals – the precision of the local information, relative to its surround. In practice, this could lead to an increase of fast correlations among clusters of granule cells that are excited by common MFs, and a simultaneous decrease of slower correlations across competing patches of granular layer – replicating observations in structures that share a similar geometry, like the olfactory bulb [24].

This contextual modulation of granule cell excitability relies on spatial constraints of information driving Golgi and granule cell populations, which in turn depend on different anatomical properties of the network. MFs show substantial anisotropic divergence in the granular layer [25], which enables integration of various sources of information at the level of single granule cells, but prevents the emergence of ordered, neocortical-like receptive fields. As a consequence, fast correlations among Golgi cells (and inhibited clusters of granule cells) sharing MF input might be more evident within distributed, scattered groups of cells [26].

Another important anatomical property is the presence of millimeter-long granule-Golgi cell connections mediated by parallel fibres [27]. Parallel fibres have been linked to extended oscillations in the granular layer during rest [28], possibly setting a global pace for network computations and dynamics. Notably, these connections appear to be qualitatively different from local contacts made by ascending granule cell axons onto Golgi cells, which resemble more the faster and stronger MF-Golgi cell synapses [29, 30]. It follows that upon localised activation of MF terminals, parallel fibres might preferentially contribute to slow correlation of granule cells across the transverse axis [31], while ascending axons precisely entrain spiking of surrounding Golgi cells. Furthermore, the

existence of electrical connections among Golgi cells further increases their sensitivity to temporal coincidence of local excitation, enhancing synchrony or alternatively asynchrony in and between granule cell clusters [32, 33, 34]. Therefore, different degrees of correlations might coexist in the granular layer, following properties of MF input and connectivity structure within the network, which might result in balanced dynamics of excitation and inhibition reflecting the statistics (precision) of information encoded.

Finally, precision-weighting for state estimation does not depend solely on properties intrinsic to the inputs, but also on selective mechanisms modulating states of the network. Analogously, Golgi cells are both driven by the same MF inputs that elicit activity in granule cells, and are influenced by neural components located within or external to the cerebellum. *In vivo*, the granular layer is characterised by endogenous activity due to spontaneous firing of MFs and Golgi cells [82, 28, 79]; this autonomous state affects the evoked response elicited by a stimulus, and is itself under the control of various mechanisms. In particular, within the cerebellar cortex, climbing fibres, Lugaro cells and Purkinje cells all directly or indirectly modulate Golgi cell activity. [83, 84, 85]. From cerebellar nuclei instead, excitatory neurons give rise to MF collaterals innervating glomeruli [86], and inhibitory neurons selectively contact Golgi cells through long-range axons [87]. Moreover, Golgi cells are also sensitive to a variety of neuromodulators including serotonin [88] and noradrenaline [89], which exert opposing actions upon granular layer excitability. Clearly, these sources of input exert very different effects on information processing, and their specific role is still unresolved; nevertheless, this intricate circuit highlights the importance of properly tuning inhibition in the granular layer in order to contextualise incoming information. This is central for **putative state estimation in the cerebellar cortex**, as it depends not only on current local observations, but also on past inference, system-wise states, and coordination with other brain structures.

In conclusion, there appear to be a variety of mechanisms that could inform the granular layer about precision of MF input, irrespectively of the extremely diversified nature of those input. These mechanisms condition granule cell excitation through Golgi cell inhibition, which constitutes the unique local feedback of the network. In this sense, Golgi cells emerge as a crucial hub for precise state estimation in the cerebellar cortex (Figure 2).”

We also mention variability in Golgi-granule cell synapses in the discussion (lines 270-285):

“Testing these ideas requires both a theoretical and experimental efforts. Here we have assumed a general principle, namely, that precision in neural representations should affect their propagation across the different stages of inference – by tuning population gain. However, future work should aim at investigate the exact nature of this probabilistic encoding throughout the cerebellar circuit. From the experimental side, testing these ideas requires tracking and manipulation of spatiotemporal properties of excitation-inhibition balance in the granular layer, as MF input is transformed into parallel fibre output. The exact shape of ensuing activity patterns depends on many factors, including kinetics variability at the Golgi-granule cell synapses [68], which might result from plastic mechanisms in the granular layer [69]. Previous works have examined the consequences of altering excitation levels in this network, showing a direct link to motor impairment, including tremors, ataxia and reduced reflex adaptation [67, 70, 71]. Interestingly, Golgi cell ablation alters the spatiotemporal patterns of activity in the granule cell population without necessarily producing overexcitation, as the result of compensatory mechanisms such as reduced NMDA activity

[72]. This highlights the importance of fine-grained granule cell activity patterns for downstream computations [71] – here argued to be the result of Golgi cell-mediated precision-weighting at the first stage of state estimation in the cerebellar cortex. Ultimately, technical advances in the field [73, 74] will make possible to verify or not these ideas.”

10) It is unclear why the adaptive filter theory should be completely at odds with the precision-weighting. Instead of assuming a stable or uniform set of temporal basis functions, inhibition can change the dynamical repertoire of the network, that constrain or shape estimation at the output layer. Further, the experimental studies all report granule cell responses to external sensory stimuli, without any learning or predictive task component.

Thanks for the suggestion. Unfortunately, because of limited word count, we had to omit the section about cerebellar models. As for learning or predictive task components in granule cell activity, this has been seen in recent papers and is likely inherited from mossy fibre stimulation. We address this issue in the last paragraph of the section “Precision in state estimation” (lines 93-106):

“The cerebellar cortex receives input via MFs from virtually every part of the brain. This input is rich, encompassing multiple sensory and motor modalities [5, 6, 7, 8, 9] as well as cognitive domains [10]. Moreover, its nature can be both predictive (e.g. anticipatory reward-related signals) and postdictive (e.g. sensory feedback) [11, 12, 13], encompassing the entire period of movement execution (see e.g. [12]). Consequently, at any given time a huge amount of information can potentially be transmitted to the granular layer via MFs. However, only a fraction of this information is likely to be relevant at any given moment in time; this fact intimates the necessity for the cerebellar cortex to select or prioritise some and not other sources of input, so that only information that is relevant in a particular behavioural context can affect state estimation. For example, while engaged in a visuomotor task, postsynaptic responses to MFs conveying confounding auditory signals might be dampened. **With respect to the cerebellar cortex, precision encoded in various extracerebellar regions must be translated and implemented in a common way within the granule cell population, in a manner which is instrumental for state estimation, that is, causing downstream layers to appropriately respond to encoded precision. Accordingly, inhibition in the input layer of the cerebellum appears capable of operating these fundamental operations.**”

References

- [1] Y. Kubota, “Untangling GABAergic wiring in the cortical microcircuit,” *Current Opinion in Neurobiology*, vol. 26, p. 714, 2014.
- [2] H. Markram, M. Toledo-Rodriguez, Y. Wang, A. Gupta, G. Silberberg, and C. Wu, “Interneurons of the neocortical inhibitory system,” *Nature Reviews Neuroscience*, vol. 5, no. 10, p. 793807, 2004.
- [3] M. Hamann, D. J. Rossi, and D. Attwell, “Tonic and spillover inhibition of granule cells control information flow through cerebellar cortex,” *Neuron*, vol. 33, no. 4, p. 625633, 2002.

- [4] S. J. Mitchell and R. Silver, “Shunting inhibition modulates neuronal gain during synaptic excitation,” *Neuron*, vol. 38, no. 3, pp. 433 – 445, 2003.
- [5] R. S. Snider and A. Stowell, “Receiving areas of the tactile, auditory, and visual systems in the cerebellum,” *Journal of Neurophysiology*, vol. 7, p. 331357, Jan 1944.
- [6] N. Sobel, V. Prabhakaran, C. A. Hartley, J. E. Desmond, Z. Zhao, G. H. Glover, J. D. Gabrieli, and E. V. Sullivan, “Odorant-induced and sniff-induced activation in the cerebellum of the human,” *The Journal of Neuroscience*, vol. 18, p. 89909001, Jan 1998.
- [7] C.-C. Huang, K. Sugino, Y. Shima, C. Guo, S. Bai, B. D. Mensh, S. B. Nelson, and A. W. Hantman, “Convergence of pontine and proprioceptive streams onto multimodal cerebellar granule cells,” *eLife*, vol. 2, 2013.
- [8] F. P. Chabrol, A. Arenz, M. T. Wiechert, T. W. Margrie, and D. A. Digregorio, “Synaptic diversity enables temporal coding of coincident multisensory inputs in single neurons,” *Nature Neuroscience*, vol. 18, no. 5, p. 718727, 2015.
- [9] T. Ishikawa, M. Shimuta, and M. Husser, “Multimodal sensory integration in single cerebellar granule cells in vivo,” *eLife*, vol. 4, 2015.
- [10] M. J. Wagner and L. Luo, “Neocortex-cerebellum circuits for cognitive processing,” *Trends in Neurosciences*, vol. 43, no. 1, p. 4254, 2020.
- [11] A. Giovannucci, A. Badura, B. Deverett, F. Najafi, T. D. Pereira, Z. Gao, I. Ozden, A. D. Kloth, E. Pnevmatikakis, L. Paninski, C. I. De Zeeuw, J. F. Medina, and S. S.-H. Wang, “Cerebellar granule cells acquire a widespread predictive feedback signal during motor learning,” *Nature Neuroscience*, vol. 20, pp. 727 EP –, Mar 2017.
- [12] M. J. Wagner, T. H. Kim, J. Savall, M. J. Schnitzer, and L. Luo, “Cerebellar granule cells encode the expectation of reward,” *Nature*, vol. 544, no. 7648, p. 96100, 2017.
- [13] L. S. Popa and T. J. Ebner, “Cerebellum, predictions and errors,” *Frontiers in Cellular Neuroscience*, vol. 12, 2019.
- [14] J. C. Eccles, M. Ito, and S. János, *The cerebellum as a neuronal machine*. Springer, 1967.
- [15] M. Ito, M. Itō, and J. Eccles, *The Cerebellum and Neural Control*. Raven Press, 1984.
- [16] M. Palkovits, P. Magyar, and J. Szentgothai, “Quantitative histological analysis of the cerebellar cortex in the cat,” *Brain Research*, vol. 32, no. 1, p. 1530, 1971.
- [17] G. Billings, E. Piasini, A. Lrincz, Z. Nusser, and R. A. Silver, “Network structure within the cerebellar input layer enables lossless sparse encoding,” *Neuron*, vol. 83, no. 4, p. 960974, 2014.
- [18] R. L. Jakab and J. HáMori, “Quantitative morphology and synaptology of cerebellar glomeruli in the rat,” *Anatomy and Embryology*, vol. 179, no. 1, p. 8188, 1988.
- [19] D. McNamee and D. M. Wolpert, “Internal models in biological control,” *Annual Review of Control, Robotics, and Autonomous Systems*, vol. 2, p. 339364, Mar 2019.

- [20] R. P. Rao, “Bayesian inference and attentional modulation in the visual cortex,” *NeuroReport*, vol. 16, no. 16, p. 18431848, 2005.
- [21] H. Feldman and K. Friston, “Attention, uncertainty, and free-energy,” *Frontiers in Human Neuroscience*, vol. 4, p. 215, 2010.
- [22] A. J. Yu and P. Dayan, “Inference, attention, and decision in a bayesian neural architecture,” in *Advances in Neural Information Processing Systems 17* (L. K. Saul, Y. Weiss, and L. Bottou, eds.), pp. 1577–1584, MIT Press, 2005.
- [23] E. D’Angelo, S. Solinas, J. Mapelli, D. Gandolfi, L. Mapelli, and F. Prestori, “The cerebellar Golgi cell and spatiotemporal organization of granular layer activity,” *Frontiers in Neural Circuits*, vol. 7, 2013.
- [24] S. Giridhar, B. Doiron, and N. N. Urban, “Timescale-dependent shaping of correlation by olfactory bulb lateral inhibition,” *Proceedings of the National Academy of Sciences*, vol. 108, no. 14, pp. 5843–5848, 2011.
- [25] F. Sultan, “Distribution of mossy fibre rosettes in the cerebellum of cat and mice: evidence for a parasagittal organization at the single fibre level,” *European Journal of Neuroscience*, vol. 13, no. 11, pp. 2123–2130, 2001.
- [26] H. Gurnani and R. A. Silver, “Coordination of inhibitory Golgi cell population activity in the cerebellar cortex.” *Cosyne Abstracts 2019*, 2019.
- [27] R. J. Harvey and R. M. A. Napper, “Quantitative study of granule and Purkinje cells in the cerebellar cortex of the rat,” *Journal of Comparative Neurology*, vol. 274, no. 2, pp. 151–157, 1988.
- [28] B. P. Vos, R. Maex, A. Volny-Luraghi, and E. D. Schutter, “Parallel fibers synchronize spontaneous activity in cerebellar Golgi cells,” *Journal of Neuroscience*, vol. 19, no. 11, pp. RC6–RC6, 1999.
- [29] D. Watanabe and S. Nakanishi, “mGluR2 postsynaptically senses granule cell inputs at Golgi cell synapses,” *Neuron*, vol. 39, no. 5, pp. 821 – 829, 2003.
- [30] E. Cesana, K. Pietrajtis, C. Bidoret, P. Isope, E. D’Angelo, S. Dieudonne, and L. Forti, “Granule cell ascending axon excitatory synapses onto Golgi cells implement a potent feedback circuit in the cerebellar granular layer,” *Journal of Neuroscience*, vol. 33, no. 30, p. 1243012446, 2013.
- [31] S. K. Sudhakar, S. Hong, I. Raikov, R. Publio, C. Lang, T. Close, D. Guo, M. Negrello, and E. De Schutter, “Spatiotemporal network coding of physiological mossy fiber inputs by the cerebellar granular layer,” *PLOS Computational Biology*, vol. 13, pp. 1–35, 09 2017.
- [32] G. P. Dugu, N. Brunel, V. Hakim, E. Schwartz, M. Chat, M. Lvesque, R. Courtemanche, C. Lna, and S. Dieudonn, “Electrical coupling mediates tunable low-frequency oscillations and resonance in the cerebellar Golgi cell network,” *Neuron*, vol. 61, no. 1, pp. 126 – 139, 2009.

- [33] K. Vervaeke, A. Lrincz, P. Gleeson, M. Farinella, Z. Nusser, and R. A. Silver, “Rapid desynchronization of an electrically coupled interneuron network with sparse excitatory synaptic input,” *Neuron*, vol. 67, no. 3, p. 435451, 2010.
- [34] I. vanWelle, A. Roth, S. S. Ho, S. Komai, and M. Husser, “Conditional spike transmission mediated by electrical coupling ensures millisecond precision-correlated activity among interneurons *in vivo*,” *Neuron*, vol. 90, no. 4, pp. 810 – 823, 2016.
- [35] D. C. Knill and A. Pouget, “The bayesian brain: the role of uncertainty in neural coding and computation,” *Trends in Neurosciences*, vol. 27, no. 12, p. 712719, 2004.
- [36] K. P. Krording and D. M. Wolpert, “Bayesian integration in sensorimotor learning,” *Nature*, vol. 427, no. 6971, p. 244247, 2004.
- [37] J. Fiser, P. Berkes, G. Orbán, and M. Lengyel, “Statistically optimal perception and learning: from behavior to neural representations,” *Trends in Cognitive Sciences*, vol. 14, no. 3, p. 119130, 2010.
- [38] M. G. Paulin, “A model of the role of the cerebellum in tracking and controlling movements,” *Human Movement Science*, vol. 12, no. 1, pp. 5 – 16, 1993.
- [39] X. Liu, E. Robertson, and R. C. Miall, “Neuronal activity related to the visual representation of arm movements in the lateral cerebellar cortex,” *Journal of Neurophysiology*, vol. 89, p. 12231237, Jan 2003.
- [40] N. L. Cerminara, R. Apps, and D. E. Marple-Horvat, “An internal model of a moving visual target in the lateral cerebellum,” *The Journal of Physiology*, vol. 587, no. 2, p. 429442, 2009.
- [41] D. M. Wolpert, R. Miall, and M. Kawato, “Internal models in the cerebellum,” *Trends in Cognitive Sciences*, vol. 2, no. 9, p. 338347, 1998.
- [42] M. Ito, “Cerebellar circuitry as a neuronal machine,” *Progress in Neurobiology*, vol. 78, no. 3, pp. 272 – 303, 2006. *The Contributions of John Carew Eccles to Contemporary Neuroscience*.
- [43] J. X. Brooks, J. Carriot, and K. E. Cullen, “Learning to expect the unexpected: rapid updating in primate cerebellum during voluntary self-motion,” *Nature Neuroscience*, vol. 18, pp. 1310–1317, Sep 2015.
- [44] M. Ito, “Control of mental activities by internal models in the cerebellum,” *Nature Reviews Neuroscience*, vol. 9, no. 4, p. 304313, 2008.
- [45] M. G. Paulin, “Evolution of the cerebellum as a neuronal machine for bayesian state estimation,” *Journal of Neural Engineering*, vol. 2, pp. S219–S234, aug 2005.
- [46] W. J. Ma, J. M. Beck, P. E. Latham, and A. Pouget, “Bayesian inference with probabilistic population codes,” *Nature Neuroscience*, vol. 9, pp. 1432–1438, Nov 2006.
- [47] G. Orbán, P. Berkes, J. Fiser, and M. Lengyel, “Neural variability and sampling-based probabilistic representations in the visual cortex,” *Neuron*, vol. 92, no. 2, pp. 530 – 543, 2016.
- [48] S. Chen, G. J. Augustine, and P. Chadderton, “Serial processing of kinematic signals by cerebellar circuitry during voluntary whisking,” *Nature Communications*, vol. 8, Oct 2017.

- [49] M. I. Becker and A. L. Person, “Cerebellar control of reach kinematics for endpoint precision,” *Neuron*, vol. 103, no. 2, pp. 335 – 348.e5, 2019.
- [50] R. M. Kelly and P. L. Strick, “Cerebellar loops with motor cortex and prefrontal cortex of a nonhuman primate,” *Journal of Neuroscience*, vol. 23, no. 23, pp. 8432–8444, 2003.
- [51] R. D. Proville, M. Spolidoro, N. Guyon, G. P. Dugué, F. Selimi, P. Isope, D. Popa, and C. Léna, “Cerebellum involvement in cortical sensorimotor circuits for the control of voluntary movements,” *Nature Neuroscience*, vol. 17, pp. 1233–1239, Sep 2014.
- [52] Z. Gao, C. Davis, A. M. Thomas, M. N. Economo, A. M. Abrego, K. Svoboda, C. I. De Zeeuw, and N. Li, “A cortico-cerebellar loop for motor planning,” *Nature*, vol. 563, pp. 113–116, Nov 2018.
- [53] F. P. Chabrol, A. Blot, and T. D. Mrsic-Flogel, “Cerebellar contribution to preparatory activity in motor neocortex,” *Neuron*, vol. 103, no. 3, pp. 506 – 519.e4, 2019.
- [54] M. Heilbron and M. Chait, “Great expectations: Is there evidence for predictive coding in auditory cortex?,” *Neuroscience*, vol. 389, pp. 54 – 73, 2018. Sensory Sequence Processing in the Brain.
- [55] P. Fries, T. Womelsdorf, R. Oostenveld, and R. Desimone, “The effects of visual stimulation and selective visual attention on rhythmic neuronal synchronization in macaque area V4,” *Journal of Neuroscience*, vol. 28, no. 18, p. 48234835, 2008.
- [56] T. Akam and D. M. Kullmann, “Oscillations and filtering networks support flexible routing of information,” *Neuron*, vol. 67, no. 2, pp. 308 – 320, 2010.
- [57] C. M. Lewis, J. Ni, T. Wunderle, P. Jendritza, I. Diester, and P. Fries, “Gamma-rhythmic input causes spike output,” *bioRxiv*, 2020.
- [58] M. Farrant and Z. Nusser, “Variations on an inhibitory theme: phasic and tonic activation of GABA_A receptors,” *Nature Reviews Neuroscience*, vol. 6, no. 3, p. 215229, 2005.
- [59] D. J. Rossi, M. Hamann, and D. Attwell, “Multiple modes of GABAergic inhibition of rat cerebellar granule cells,” *The Journal of Physiology*, vol. 548, no. 1, p. 97110, 2003.
- [60] S. Lee, B.-E. Yoon, K. Berglund, S.-J. Oh, H. Park, H.-S. Shin, G. J. Augustine, and C. J. Lee, “Channel-mediated tonic GABA release from glia,” *Science*, vol. 330, no. 6005, pp. 790–796, 2010.
- [61] D. A. Digregorio, Z. Nusser, and R. Silver, “Spillover of glutamate onto synaptic AMPA receptors enhances fast transmission at a cerebellar synapse,” *Neuron*, vol. 35, no. 3, p. 521533, 2002.
- [62] T. A. Nielsen, D. A. Digregorio, and R. Silver, “Modulation of glutamate mobility reveals the mechanism underlying slow-rising AMPAR EPSCs and the diffusion coefficient in the synaptic cleft,” *Neuron*, vol. 42, no. 5, p. 757771, 2004.
- [63] D. J. Rossi and M. Hamann, “Spillover-mediated transmission at inhibitory synapses promoted by high affinity $\alpha 6$ subunit GABA_A receptors and glomerular geometry,” *Neuron*, vol. 20, no. 4, p. 783795, 1998.

- [64] S. G. Brickley, V. Revilla, S. G. Cull-Candy, W. Wisden, and M. Farrant, “Adaptive regulation of neuronal excitability by a voltage-independent potassium conductance,” *Nature*, vol. 409, no. 6816, p. 8892, 2001.
- [65] P. Chadderton, T. W. Margrie, and M. Häusser, “Integration of quanta in cerebellar granule cells during sensory processing,” *Nature*, vol. 428, no. 6985, pp. 856–860, 2004.
- [66] I. Duguid, T. Branco, M. London, P. Chadderton, and M. Häusser, “Tonic inhibition enhances fidelity of sensory information transmission in the cerebellar cortex,” *Journal of Neuroscience*, vol. 32, no. 32, pp. 11132–11143, 2012.
- [67] H. J. Hanchar, P. D. Dodson, R. W. Olsen, T. S. Otis, and M. Wallner, “Alcohol-induced motor impairment caused by increased extrasynaptic gabaa receptor activity,” *Nature Neuroscience*, vol. 8, pp. 339–345, Mar 2005.
- [68] J. J. Crowley, D. Fioravante, and W. G. Regehr, “Dynamics of fast and slow inhibition from cerebellar Golgi cells allow flexible control of synaptic integration,” *Neuron*, vol. 63, no. 6, pp. 843 – 853, 2009.
- [69] C. Hansel, D. J. Linden, and E. D’Angelo, “Beyond parallel fiber LTD: the diversity of synaptic and non-synaptic plasticity in the cerebellum,” *Nature Neuroscience*, vol. 4, pp. 467–475, May 2001.
- [70] C.-S. Chiu, S. Brickley, K. Jensen, A. Southwell, S. Mckinney, S. Cull-Candy, I. Mody, and H. A. Lester, “GABA transporter deficiency causes tremor, ataxia, nervousness, and increased GABA-induced tonic conductance in cerebellum,” *Journal of Neuroscience*, vol. 25, no. 12, pp. 3234–3245, 2005.
- [71] P. Seja, M. Schonewille, G. Spitzmaul, A. Badura, I. Klein, Y. Rudhard, W. Wisden, C. A. Hbner, C. I. De Zeeuw, and T. J. Jentsch, “Raising cytosolic Cl in cerebellar granule cells affects their excitability and vestibulo-ocular learning,” *The EMBO Journal*, vol. 31, no. 5, pp. 1217–1230, 2012.
- [72] D. Watanabe, H. Inokawa, K. Hashimoto, N. Suzuki, M. Kano, R. Shigemoto, T. Hirano, K. Toyama, S. Kaneko, M. Yokoi, K. Moriyoshi, M. Suzuki, K. Kobayashi, T. Nagatsu, R. J. Kreitman, I. Pastan, and S. Nakanishi, “Ablation of cerebellar Golgi cells disrupts synaptic integration involving GABA inhibition and NMDA receptor activation in motor coordination,” *Cell*, vol. 95, pp. 17–27, Oct 1998.
- [73] V. A. Griffiths, A. M. Valera, J. Y. Lau, H. Roš, T. J. Younts, B. Marin, C. Baragli, D. Coyle, G. J. Evans, G. Konstantinou, T. Koimtzis, K. M. N. S. Nadella, S. A. Punde, P. A. Kirkby, I. H. Bianco, and R. A. Silver, “Real-time 3d movement correction for two-photon imaging in behaving animals,” *Nature Methods*, vol. 17, pp. 741–748, Jul 2020.
- [74] A. Antonini, A. Sattin, M. Moroni, S. Bovetti, C. Moretti, F. Succol, A. Forli, D. Vecchia, V. P. Rajamanickam, A. Bertoncini, S. Panzeri, C. Liberale, and T. Fellin, “Extended field-of-view ultrathin microendoscopes for high-resolution two-photon imaging with minimal invasiveness in awake mice,” *bioRxiv*, 2020.

- [75] R. T. Kanichay and R. A. Silver, “Synaptic and cellular properties of the feedforward inhibitory circuit within the input layer of the cerebellar cortex,” *Journal of Neuroscience*, vol. 28, p. 89558967, Mar 2008.
- [76] I. Duguid, T. Branco, P. Chadderton, C. Arlt, K. Powell, and M. Häusser, “Control of cerebellar granule cell output by sensory-evoked Golgi cell inhibition,” *Proceedings of the National Academy of Sciences*, vol. 112, no. 42, pp. 13099–13104, 2015.
- [77] H. Jörntell and C.-F. Ekerot, “Properties of somatosensory synaptic integration in cerebellar granule cells in vivo,” *Journal of Neuroscience*, vol. 26, p. 1178611797, Aug 2006.
- [78] E. A. Rancz, T. Ishikawa, I. Duguid, P. Chadderton, S. Mahon, and M. Häusser, “High-fidelity transmission of sensory information by single cerebellar mossy fibre boutons,” *Nature*, vol. 450, pp. 1245 EP –, Dec 2007.
- [79] A. Arenz, R. A. Silver, A. T. Schaefer, and T. W. Margrie, “The contribution of single synapses to sensory representation in vivo,” *Science*, vol. 321, no. 5891, p. 977980, 2008.
- [80] F. Bengtsson and H. Jörntell, “Sensory transmission in cerebellar granule cells relies on similarly coded mossy fiber inputs,” *Proceedings of the National Academy of Sciences*, vol. 106, no. 7, p. 23892394, 2009.
- [81] K. Powell, A. Mathy, I. Duguid, and M. Husser, “Synaptic representation of locomotion in single cerebellar granule cells,” *eLife*, vol. 4, 2015.
- [82] P. L. van Kan, A. R. Gibson, and J. C. Houk, “Movement-related inputs to intermediate cerebellum of the monkey,” *Journal of Neurophysiology*, vol. 69, no. 1, pp. 74–94, 1993. PMID: 8433135.
- [83] A. K. Nietz, J. H. Vaden, L. T. Coddington, L. Overstreet-Wadiche, and J. I. Wadiche, “Non-synaptic signaling from cerebellar climbing fibers modulates Golgi cell activity,” *eLife*, vol. 6, 2017.
- [84] S. Dieudonn and A. Dumoulin, “Serotonin-driven long-range inhibitory connections in the cerebellar cortex,” *The Journal of Neuroscience*, vol. 20, p. 18371848, Jan 2000.
- [85] L. Witter, S. Rudolph, R. T. Pressler, S. I. Lahlaf, and W. G. Regehr, “Purkinje cell collaterals enable output signals from the cerebellar cortex to feed back to Purkinje cells and interneurons,” *Neuron*, vol. 91, no. 2, p. 312319, 2016.
- [86] B. D. Houck and A. L. Person, “Cerebellar premotor output neurons collateralize to innervate the cerebellar cortex,” *Journal of Comparative Neurology*, vol. 523, p. 22542271, Dec 2015.
- [87] L. Ankri, Z. Husson, K. Pietrajtis, R. Proville, C. Lna, Y. Yarom, S. Dieudonn, and M. Y. Uusisaari, “A novel inhibitory nucleo-cortical circuit controls cerebellar Golgi cell activity,” *eLife*, vol. 4, p. e06262, may 2015.
- [88] E. Fleming and C. Hull, “Serotonin regulates dynamics of cerebellar granule cell activity by modulating tonic inhibition,” *Journal of Neurophysiology*, vol. 121, p. 105114, Jan 2019.
- [89] F. Lanore, J. S. Rothman, D. Coyle, and R. A. Silver, “Norepinephrine controls the gain of the inhibitory circuit in the cerebellar input layer,” *bioRxiv*, 2019.

Appendix B

Response to referees

2nd referee:

Comments to the Author(s) The authors have done a good job in revising the article, which now acknowledges previous work on this subject and incorporates the latest research. There are however a few minor typos to correct.

Ln 144 F-I not defined.

Ln 165 '6' should be subscript

Ln 205 Huang et al., eLife 2013 could be added here.

Ln 327 HaMori. The 'M' should not be capitalized.

Thanks for spotting these typos. We amended the text as follow:

Ln 144 F-I not defined.

“As a result, Golgi cells can set the excitability or responsiveness of granule cells, approximated by the operative point (position and slope) of their **F-I (frequency-current) curve**, controlling propagation of MF activity within the cerebellar circuit.”

Ln 165 '6' should be subscript

“In these compartments, ambient concentrations of GABA are sufficient to persistently activate high-affinity $\alpha_6\delta$ -subunit containing GABAA receptors”

Ln 205 Huang et al., eLife 2013 could be added here.

“MFs show substantial anisotropic divergence in the granular layer [1, 2], which enables integration of various sources of information at the level of single granule cells, but prevents the emergence of ordered, neocortical-like receptive fields.”

Ln 327 HaMori. The 'M' should not be capitalized.

[3]

References

- [1] F. Sultan, “Distribution of mossy fibre rosettes in the cerebellum of cat and mice: evidence for a parasagittal organization at the single fibre level,” *European Journal of Neuroscience*, vol. 13, no. 11, pp. 2123–2130, 2001.
- [2] C.-C. Huang, K. Sugino, Y. Shima, C. Guo, S. Bai, B. D. Mensh, S. B. Nelson, and A. W. Hantman, “Convergence of pontine and proprioceptive streams onto multimodal cerebellar granule cells,” *eLife*, vol. 2, 2013.
- [3] R. L. Jakab and J. Hámori, “Quantitative morphology and synaptology of cerebellar glomeruli in the rat,” *Anatomy and Embryology*, vol. 179, no. 1, p. 8188, 1988.